# Can system dynamics explain long-term hydrological behaviors? The role of endogenous linking structure

Xinyao Zhou[1], Zhuping Sheng[2], Kiril Manevski[3,4,5], Rongtian Zhao[6,7], Qingzhou Zhang[8], Yanmin Yang[1], Shumin Han[1], Jinghong Liu[1,9], Yonghui Yang[1,4,9]

[1]Key Laboratory of Agricultural Water Resources, Hebei Laboratory of Agricultural Water-Saving, Center for Agricultural Resources Research, Institute of Genetics and Developmental Biology, Chinese Academy of Sciences, Shijiazhuang 050021, China
[2]Department of Civil Engineering, Morgan State University, Baltimore, MD 21251, USA
[3]Department of Agroecology, Aarhus University, Tjele 8830, Denmark
[4]Sino-Danish College, University of Chinese Academy of Sciences, Yanqihu Campus, Beijing 101408, China
[5]iClimate – Aarhus University Interdisciplinary Center for Climate Change, Department of Environmental Science, Roskilde 4000, Denmark
[6]State Key Laboratory of Resources and Environmental Information System, Institute of Geographic Sciences and Natural Resources Research, Chinese Academy of Sciences, Beijing 100101, China
[7]College of Resources and Environment, University of Chinese Academy of Sciences, Beijing 100049, China
[8]Land Resources Exploration Center of Hebei Bureau of Geology and Mineral Exploration and Development (Hebei Mine and Geological Disaster Emergency Rescue Center), Shijiazhuang 050081, China
[9]University of Chinese Academy of Sciences, Beijing 100190, China

*Correspondence to*: Xinyao Zhou (zhouxy@sjziam.ac.cn) and Yonghui Yang (yonghui.yang@sjziam.ac.cn)

**Abstract.** Hydrological models with the conceptual tipping bucket and the process-based evapotranspiration formulations are the most common tools in hydrology. However, these models consistently fail to replicate long-term and slow dynamics of a hydrological system, indicating the need for model augmentation and a shift in formulation approach. This study employed an entirely different approach – system dynamics – towards more realistic replication of the observed slow hydrological behaviors at inter-annual and inter-decadal scales. Using the headwaters of Baiyang Lake in China as a case study, the endogenous linking structure of the hydrological system was gradually unravelled from 1982 to 2015 through wavelet analysis, Granger's causality test, and system dynamics model. The wavelet analysis and the Granger's causality test identified a negatively correlated and bidirectional causal relationship between actual evapotranspiration and catchment water storage change across distinct climatic periodicities, and the system dynamics approach suggested a combined structure of a vegetation reinforcing feedback and a soil water-vegetation balancing feedback in the hydrological system. The system dynamics' structure successfully captured the slow hydrological behaviors under both natural and human-intervention scenarios, demonstrating a self-sustained oscillation arising within the system's boundary. Our results showed that the interaction between the vegetation structure and the soil-bound water dominates the hydrological process at an inter-annual scale, while the interaction between the climatic oscillation and the soil water holding capacity dominates the hydrological process at an inter-decadal scale. Conventional hydrological models, which typically employ physiological-based evapotranspiration formulations and assume invariable soil characteristics, ignore vegetation structure change at the inter-

annual scale and soil water holding capacity change at the inter-decadal scale, leading to failure in predicting the observed long-term hydrological behaviors. The system dynamics model is in its early stage with applications primarily confined to water-stressed regions and long-term scales. However, the novel insights proposed in our study, including the different hierarchies corresponding to distinct mechanisms and time scales, and the endogenous linking structure among stocks being more important driver of the hydrological behaviors, offer potential solutions for better understanding a hydrological system and guidelines for improving configuration and performance of conventional hydrological models.

**Short Summary.** Conventional hydrological models erratically replicate slow hydrological dynamics, necessitating model modification and paradigm shift in hydrological science. The system dynamics approach successfully explains patterns of slow hydrological behaviors at inter-annual and decadal scales by dividing a hydrological system into different hierarchies and building endogenous linking structure among stocks. In spite of the simplicity, it holds potential to integrate hydrological behaviors across scales.

## 1 Introduction

Hydrological models are scientific tools for describing and predicting the processes of the water cycle under the current and the future climate. In the simplest concept, a "bucket" is used to represent the water storage of a catchment, which fills by rain and empties by evaporation, transpiration and streamflow (Fowler et al., 2020). In spite of the success of this concept in understanding the physical processes and revealing the physical parameters involved in the hydrological events, such conventional hydrological models have performed poorly in simulating the long-term and the slowly-variable hydrological system observed in many regions across the world, e.g., the inter-annual cycles of catchment water storage (Creutzfeldt et al., 2012; Hulsman et al., 2021; Chang and Niu, 2023), and the multi-decadal decline of runoff (Chen et al., 2016; Peterson et al., 2021). Consistent inaccuracies of the conventional hydrological models in replicating slow hydrological behaviors suggest a lack of key mechanisms involved in the hydrological system. Alongside structural deficiencies (Fowler et al., 2020; Bouaziz et al., 2021), reasons for erratic model performance also involve data errors (Kuczera et al., 2010; Hulsman et al., 2021), model structural deficiencies (Fowler et al., 2020; Bouaziz et al., 2021), poor parameterization (Fowler et al., 2016, 2018), or their combination. However, model structural deficiency likely plays the key role in most cases of poor performance (Fowler et al., 2020).

From a hydrological perspective, two main solutions have been proposed to improve the conventional hydrological model structure. The first introduces a "bottomless bucket" to avoid the "minimum possible storage" limit to the bucket, which empties the bucket on a seasonal basis and restricts the accumulation of water deficit (Fowler et al., 2020). The "bottomless bucket" configuration better tracks a long-term decline in soil moisture seen, for instance, in Australia during the 13-year "Millennium" drought desiccating catchments slowly and gradually (Fowler et al., 2021). On the other hand, groundwater usually responds slowly to rainfall variability and the lag time scales are mediated by the hydrogeology of the aquifers and

the soil physical characteristics (Hughes et al., 2012; Markovich et al., 2016). Thus, it has also been suggested that inter-basin groundwater flow should be incorporated in hydrological models through a new deeper groundwater reservoir to allow

models to better reproduce the long-term storage fluctuations (Bouaziz et al., 2018; Hulsman et al., 2021). Although some aspects are improved by these reconfigurations or modifications, other aspects of the hydrology behaviors are still not captured or reflected well, for instance those related to the effect of the terrestrial vegetation on the long-term and slow dynamics in hydrological models (Fowler et al., 2021).

The system dynamics approach assumes that ample time is required for a system to undergo changes (Meadows, 2008) and

is thus a different approach to describe and interpret the mechanism of the ubiquitously long-term and slow variations of the hydrological system. Different from the conventional viewpoint postulating that the dynamics of a system are primarily driven by exogenous variables, the system dynamics approach seeks explanations of endogenous structure for the often complex, difficult-to-understand dynamics and "behaviors" (Forrester, 1968). This is because, on one hand, stocks generally change slowly, thus can act as delays, lags, buffers or shock absorbers in the system (Meadows, 2008). This characteristic is

also prominent in hydrological system and has been built in conventional hydrological models using various stocks (soil moisture/groundwater/lakes) leading to description of some slow hydrological dynamics (O'Connell et al., 2016). On the other hand, stocks are complexly linked with other even competing feedback loops operating simultaneously, creating counteracting and compensating pressures in response to exogenous disturbances (Richardson, 2020). Therefore, exogenous drivers might not be able to explain and anticipate a system's main pattern of behaviors, as the cause and the effect are often

distant in time and space in dynamically complex systems (Davis, 2003). However, the complexity of linking structure among stocks has never been well recognized in conventional hydrological models, which often link these stocks simply and sequentially. The fact that stocks usually operate at different time scales further increase the complexity of linking structure among stocks in a hydrological system. In spite of the wide application of system dynamics in different areas (Wiener, 1948; Zera, 2002; Hofkirchner and Schafranek, 2011; Seth and Bayne, 2022), including land use dynamics (Lauf et al., 2012) and

water management (Bai et al., 2021; Simonovic, 2020) in which natural system only plays a minor role, to our knowledge, the concept has rarely been used in hydrology under natural conditions. System dynamics approach may thus open a new avenue to understand the mechanisms of long-term and slow dynamics of hydrological system.

By means of the system dynamics approach, the main aim of this paper was two-fold, to 1) seek the explanation of endogenous linking structure among stocks for the long-term and slow hydrological dynamics with a case study, and 2) test

the ability of the endogenous linking structure for (re)producing and predicting long-term and slow hydrological dynamics. To make the discussion clearer, the "long-term" is constrained to the following. As a complex system, a hydrological system comprises multiple hierarchies. Each hierarchy is governed by a distinct mechanism and produces fluctuations/cycles at certain time scale. Notably, the time series yielded at each hierarchy can be considered "long-term" or "slow" compared to its predecessor. Here, we define that low hierarchy corresponds to intra-annual scale, high hierarchy corresponds to inter-annual

scale, and an even higher hierarchy corresponds to inter-decadal scale. This definition has been inspired by a previous study which result showed that a clayey-soil catchment tend to exhibit higher flow in the short term but less discharge in the long

term than its sandstone counterpart, indicating distinct underlying mechanisms to control streamflow generation at different hierarchies/time scales (Xiao et al., 2019), as well as a study which proposed a fill-spill phenomena across scales (McDonnell et al., 2021). Considering the objectives of this study and the data span, we specify "long-term" as inter-annual to inter-decadal scales.

## 2 Methodology

### 2.1 Study site description

The study area was located at Taihang Mountain in northern China and comprised of the catchments of the rivers Sha, Tang, and Juma (Fig. 1). The climate is semi-arid with continental monsoons, i.e., hot humid summers, and cold dry winters. The average annual temperature, precipitation, and potential evapotranspiration are 7.5 °C, 556 mm, and 1369 mm, respectively (Yang and Mao, 2011; Zeng et al., 2021). Rainfall is the main source of streamflow in the study area (Fu et al., 2024). The main geologic characteristics of the Taihang Mountain area are "soil and rock dual texture", with thin topsoil of 0.2-0.5 m rich in plant roots and gravel, and a thick 0.5-10 m subsoil of weathered granite gneiss with highly developed fractures (Cao et al., 2022). The soil of Taihang Mountains is primarily developed from granite, gneiss, limestone, and sandstone. Cambisols and luvisols constitute the dominant soil types, accounting for 46.86% and 15.46% of the total area, respectively (Fu et al., 2021). These soil types all have high content of sand gradation, approximately 50%, followed by silt. Clay comprises the smallest proportion of the soil content, at approximately 20% (Yang and Cao, 2021). The three rivers flow into Baiyang Lake, the largest freshwater lake in the North China Plain (Zhuang et al., 2011). Due to prolonged droughts and intensive human interference since the 1980s, the amount of inflow water to the lake has decreased significantly (Fig. S1). This has resulted in a continuous shrinkage of the water body and ecological degradation in the lake area (Han et al., 2020). Despite many efforts quantifying the impacts of climate change and human activities on the dramatic runoff reduction (Zhuang et al., 2011; Hu et al., 2012; Wang et al., 2021), there are not many studies investigated the dynamics of the catchment water storage in the study area (Chaffaut et al., 2022; Huang and Yeh, 2022).

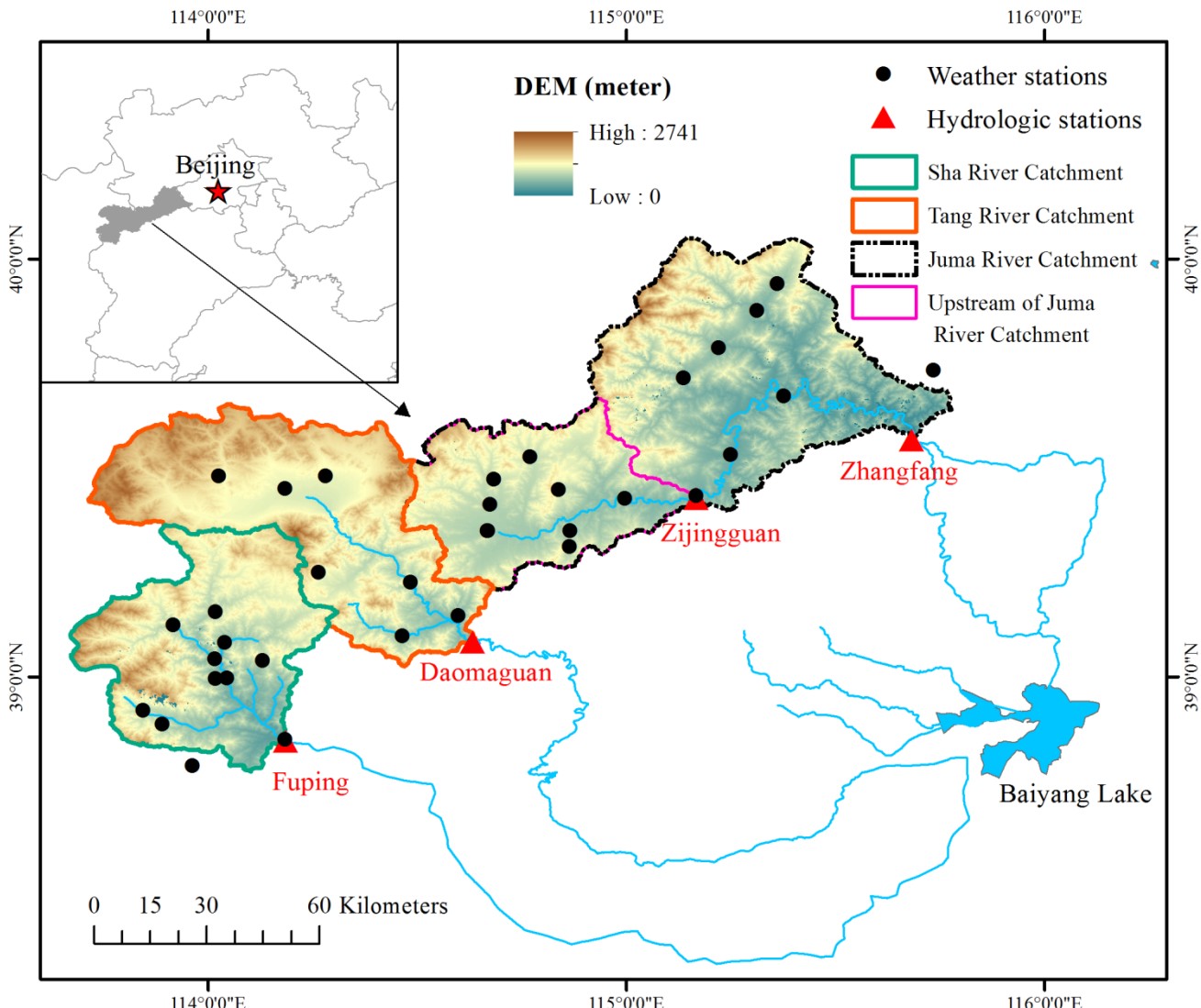

**Figure 1: The location of the study area in China at Baiyang Lake and its headwaters.**

## 2.2 Data compilation and pre-processing

Yearly streamflow, rainfall, and actual evapotranspiration data spanning the years 1982-2015 period were compiled for the catchments. Streamflow (Q) from four hydrological stations, viz, Fuping (FP), Daomaguan (DMG), Zijingguan (ZJG), and Zhangfang (ZF) were acquired from the Department of Water Resources of Hebei Province (hereafter the names of the gauge stations was used for representing the catchments; Fig. 1). FP station is located in the Sha River catchment with a drainage area of 2,160 km$^2$, whereas DMG station is located in the Tang River catchment with a drainage area of 2,704 km$^2$. ZJG and ZF are the upstream and the downstream stations of the Juma River, respectively, with corresponding drainage

areas of 1,751 and 4,737 $km^2$. The unit of streamflow data was in $m^3 s^{-1}$ in the original dataset and was converted to mm $year^{-1}$ by dividing with the area.

Precipitation data (P; mm $year^{-1}$) from 34 national weather stations within the study area were downloaded from the China Meteorological Data Sharing Service System (http://data.cma.cn/). Actual evapotranspiration (AET; mm $year^{-1}$) from 1982 to 2015 was the average of two gridded remote-sensing based AET products, Numerical Terra-dynamic Simulation Group (NTSG; University of Montata, 2023; 8 km spatial resolution) and ETWatch (8 km resolution for 1984-1989 and 1 km for 1990-2015). The AET data for the first two years were taken from NTSG and that of 1984-2015 were taken from ETWatch.

ETWatch (2023) uses the Penman-Monteith model integrated with the SEBAL and the SEBS models (Wu et al., 2008; Wu et al., 2021) and previous studies show reliable ET estimates over the Haihe Basin where the study area is located (Moiwo et al., 2011; Wu et al., 2012). The NTSG is also a Penman-Monteith model (Zhang et al., 2009, 2010). Furthermore, two alternative AET products, CR (Complementary-Relationship-based modelling, Ma et al., 2019a,b) and PEW (Proportionality hypothesis-based surface Energy-Water balance model, Fu et al., 2022a,b) were taken for cross-validation with ETWatch

and NTSG. Complementary-relationship and Priestley-Taylor approaches were employed by CR and PEW, respectively. Despite their algorithmic difference, the four AET products exhibit comparable values and common characteristics (Fig. S2). Firstly, all products successfully captured the annual AET fluctuations. Secondly, in terms of long-term trends, all products indicated a decline in AET at the end of 1990s and the beginning of 2000s. While AET data of NTSG and ETWatch showed an upward trend thereafter, AET data of CR and PEW continued to decrease during the study period. Thirdly, the magnitude

of fluctuations of P and Q reduced in the later stage. Given these observations, we conclude that the ETWatch product demonstrates reliable values and the most distinguishable pattern among the four AET products, thus deemed reliable choice for AET estimation.

Catchment water storage change ($\Delta$S; mm $year^{-1}$) was calculated by the mass balance method subtracting streamflow and actual evapotranspiration from precipitation ($\Delta$S = P - Q - AET). Detrended Total Water Storage Anomalies (TWSA; mm

$year^{-1}$) and surface Soil Moisture (SM; $m^3 m^{-3}$) data were used for comparison with $\Delta$S. The TWSA data was based on reconstructed GRACE/GRACE-FO global surface mass changes (land + ocean) from 1979 to 2020 at 0.5° spatial resolution (Li et al., 2021). Surface SM data with spatial resolution of 0.25° and temporal coverage from 1979 to 2020 were downloaded from the European Space Agency Climate Change Initiative (ESA CCI) website (Dorigo et al., 2023). This dataset had a high outperformance among satellite-based products (Ma et al., 2019). Comparison of the three datasets were

shown in Fig. S3.

Land use data for the Haihe Basin for the 1980s, 1990s, 2000s, and 2010s at 30 m spatial resolution were obtained from the ETWatch (2023) website. From 1980s to 2010s, arable land and high-coverage forest land kept decreasing, while low-coverage vegetation had greatly increased after 2000s (Fig. S4).

Future monthly rainfall ("pr" in CMIP), monthly total runoff ("mrro" in CMIP) and AET ("evspsblsoil" and "evspsblveg" in

CMIP), stemming from three scenarios (SSP126, SSP245, and SSP585) within four Global Climate Models (ACCESS-CM2, CNRM-ESM2, EC-Earth3, and GFDL-CM4), were retrieved from the CMIP6 (the phase 6 of the Coupled Model

Intercomparison Project, https://esgf-node.llnl.gov/search/cmip6/), covering the time span from 2015 to 2100. Notably, the GFDL-CM4 model lacks the SSP126 experiment, and the ACCESS-CM2 model lacks the "evspsblveg" variable. The spatial resolution varies, with 250 km for ACCESS-CM2 and CNRM-ESM2 models, and 100 km for EC-Earth2 and GFDL-CM4 models. The grid values corresponding to the study area were extracted. For ACCESS-CM2 (CNRM-ESM2), the grid value at the 62nd (83rd) row and 104th (93rd) column, corresponding to longitudes of 115.3125° (115.3125°) and latitudes of 39.375° (39.9218°), were used. Similarly, for EC-Earth3 (GFDL-CM4), the grid values at the 164th (92nd) and 165th (93rd) rows, and the 185th (130th) column, corresponding to longitudes of 114.6094° (114.375°) and 115.3125° (115.625°), and latitudes of 39.649° (39.5°), were utilized. The unit of kg m$^{-2}$ s$^{-1}$ was converted to mm month$^{-1}$ by multiplying it with 86400 seconds and 30 days. Subsequently, annual values were calculated by summing the monthly values. The annual rainfall was averaged from four GCMs and used to drive system dynamics (SD) model, generating Q and AET estimates under different SSP scenarios.

## 2.3 Formal data analysis

We propose three approaches — wavelet analysis, Granger's causality test and system dynamics. Although some information about the endogenous linking structure of hydrological system can be inferred from the wavelet analysis and the Granger's causality test, additional knowledge from multiple subjects is necessary to build the system dynamics' structure. For example, considering vegetation growth as a reinforcing feedback loop requires collective prior knowledge of plant physiology (Lian et al., 2021; Wright and Francia, 2024), resource competition theory (Craine and Dybzinski, 2013), population ecology (Snider and Brimlow, 2013), and likely other knowledge not included in this study. Therefore, the proposed three approaches run in parallel, are not sequential to help us understanding the endogenous linking structure among the stocks in the hydrological system.

### 2.3.1 Wavelet Analysis

Continuous Wavelet Transform (CWT) was employed to preliminarily identify connection patterns among variables in the time series. The wavelet transform has emerged as one of the most promising function transformation methods acting as a time and frequency localization operator (Pathak, 2009). The advantage of the wavelet transform is that it can reflect the evolution over time of non-stationary time series. Here the CWT analysis was used to generate varying coefficients that signify the similarity between the signal and mother wavelets at any specific scale base. The CWT of a function $f$ with respect to the mother wavelet $\Psi$ is defined by Pathak (2009):

$$W_f(a,b) = \frac{1}{\sqrt{|a|}} \int f(t) \Psi\left(\frac{t-b}{a}\right) dt \qquad (1)$$

where $W_f(a,b)$ is the wavelet coefficient, $a$ is the wavelet scale associated with dilation and contraction of a wavelet, and $b$ is a time index describing the location of the wavelet in time. $\Psi$ is known as "mother" wavelet because it can generate "child" wavelets by dilation and translation. Function is then processed with these child wavelets to yield wavelet coefficient (Sayood, 2012).

Prior to the analysis, the time series data of P, AET, Q and $\Delta$S were standardized by subtracting the mean and dividing the
standard deviation. Afterwards, correlation analysis was conducted based on the CWT coefficients. A contraction parameter of 8 was used in this study as the highest for the wavelet power spectrum of multi-decadal periodicities (Fig. S5).

### 2.3.2 Granger's Causality Test

Granger's Causality Test was adopted to decide causal direction of connections among variables. Correlation analysis alone can only tell how "similar" two variables are, however, correlation is neither necessary nor sufficient condition for causality
(Sugihara et al., 2012). Inferring causal direction is important for understanding how a complex system works. The Granger's Causality Test is a widely used method to investigate causality between two variables in a time series (Stokes and Purdon, 2017; Shi et al., 2022). For two time series from processes X and Y, it can be said that X does not Granger-cause Y if X, conditional on its own past, is independent of the past of Y (Banerjee et al., 2023). The typical method to test this dependency of two time series involves fitting a vector autoregressive model for X and measuring whether inclusion of Y in
that model makes the fitting error significantly lower:

$$y_i = \alpha_0 + \sum_{j=1}^{m} \alpha_j y_{i-j} + \sum_{j=1}^{m} \beta_j x_{i-j} + \varepsilon_i \tag{2}$$

where $\alpha_j$ and $\beta_j$ are the regression coefficients and $\varepsilon_i$ is the error term. The test is based on the null hypothesis:

H0: $\beta_1 = \beta_2 = ... = \beta_m = 0$

The X Granger-causes Y when the null hypothesis is rejected.
Further details on the theory behind the Granger's Causality Test and the Matlab toolbox used in this work can be found in Robert (2023).

### 2.3.3 System Dynamics approach

Causal structure is different from system dynamics' structure due to the lack of "feedback" concept in causal inference. Feedback is in the core of the system dynamics concepts (Meadows, 2008; Richardson, 2020). A feedback loop exists when
information resulting from certain actions eventually returns in some form, potentially influencing future actions (Richardson, 2020). Feedback loops can be reinforcing or balancing, the former aiming for exponential growth or accelerating collapse, as disequilibrating and destabilizing structures in systems, whereas the latter aiming for stability and resistance to change, for equilibrating or goal-seeking structures in the system (Meadows, 2008). Combined, reinforcing and balancing feedback loops can generate all manners of dynamic patterns. The inference of feedback loops should be from

prior experience. Both feedback loops coexist in the case study. The reinforcing feedback loop can be considered as the change of vegetation structure because the processes tend to self-enhance once they started. Here, vegetation structure involves non-physiologic components, e.g., change of compositions by growth and mortality (Li et al., 2023). For example, vegetation growth will create a more humid environment for the benefit of its further growth, while vegetation mortality will exacerbate deterioration and trigger a "death spiral" (Bruelheide et al., 2018). Soil moisture, on the other hand, can interrupt the self-enhancing processes of change in vegetation structure and is thus considered as a balancing feedback loop. For example, soil moisture approaching the wilting point increases the difficulties of the root to absorb water, causing slower and finally stagnant vegetation growth, and *vice versa* (Stocker et al., 2023).

How does the interplay between soil moisture and vegetation structure drive the dynamics of the hydrological system at inter-annual scale? Firstly, every balancing feedback loop has a desired goal used to compared to the actual system state. If a stock level is above or below the goal, the discrepancy between the actual and the desired levels will initiate corrective action and bring the state of the system back in line with the goal. Here the desired soil moisture was considered as a value within the range between field capacity and wilting point because at this range water can be held in soil and used for vegetation absorption. Secondly, while traditional hydrological models usually use exogenous variables to calculate AET by physical process-based models such as the Penman or the Thornthwaite models, the system dynamics approach calculated AET based on its earlier value. This is because there was an implicit vegetation structure stock behind AET. Starting with an initial vegetation structure level, the growth of new vegetation can be shown as an inflow and mortality of old vegetation as outflow, which would also drive smooth transformation of AET. Since vegetation structure stock can remember the history of changing flows and the growth/mortality of vegetation generally took several years, AET gradually changed and showed increase or decrease cycles at inter-annual scale. The equations of the dynamics of hydrological system solved for each year from 1982 to 2015 are:

$$TWS(t) = SMS(t) + GWS(t) + SWS(t) \tag{3}$$

$$SMS(t) = SMS(t-1) + \Delta s(t) * DT \tag{4}$$

$$\Delta s(t) = [(P(t) - Qr(t)] - AET(t) - RCH(t) \tag{5}$$

$$DISC(t) = [SMS(t-1) - ESMS] / DT \tag{6}$$

$$Qr(t) = [P(t) + DISC(t)] * C1 \tag{7}$$

$$AET(t) = AET(t-1) + [VEG(t) + DISC(t)] * C2 \tag{8}$$

$$RCH(t) = [P(t) * C3 + DISC(t)] * K(t) \tag{9}$$

$$Qs(t) = GWS(t-1) * C4 / DT \tag{10}$$

$$GWS(t) = GWS(t-1) + RCH(t) * DT - Qs(t) * DT - GP(t) * DT \tag{11}$$

$$SWS(t) = [Qr(t) + Qs(t)] * DT \tag{12}$$

where TWS, SMS, GWS and SWS are, respectively, total water stock, soil moisture stock, groundwater stock, and surface water stock, all in mm; $\Delta s$ is the change in water in soil moisture stock in mm year$^{-1}$; DISC in mm year$^{-1}$ is the discrepancy between actual (SMS) and desired (expected) soil moisture stock ESMS (mm); P is precipitation inflow in mm year$^{-1}$

whereas AET, RCH, Qr, and Qs are outflows indicating, respectively, actual evapotranspiration, recharge, rapid-response runoff, and slow-response runoff (mm year$^{-1}$). VEG and GP are human activity parameters for, respectively, vegetation-related activities such as reforestation, and groundwater pumpage both in mm year$^{-1}$, with their value of 0 representing no human influence. C1, C2, C3, and C4 are dimensionless coefficients with physical interpretations of respectively, proportion of impermeable area, proportion of soil bound water utilization by AET, proportion of rainfall into deeper layer, and proportion of outflowing groundwater. K is dimensionless response coefficient related to intrinsic permeability and determined by soil type only, and DT is time interval of one year in this study. The choice of the time interval is due to both the study's purpose and the fact that mean travel time of a non-groundwater-dominant catchment is around one year or slightly longer, leading to catchment soil water stock typically updates at an annual step (Sterte et al., 2021). The initial values of parameters are first obtained from literature, then calibrated against observed AET and Q.

## 3 Results

### 3.1 Preliminary Identification of Interconnection Pattern

The wavelet analysis revealed significant periodicities in the precipitation data for the study area at 7-8$^{th}$, and 14-15$^{th}$ years of the 1982-2015 period (Fig. S5). The periodicities were consistent with those of the Pacific Decadal Oscillation (PDO), which is the prime driver of inter-decadal change in summer rainfall over East China (Nalley et al., 2016; Zhu et al., 2015). Here we divide the whole study period into two distinct phases — wet and dry. Firstly, from climatic perspective, this is because, the PDO phase shifted from positive to negative mode since the late 1990s, corresponding to the decrease in rainfall in the study area (Fig. S6). Meanwhile the self-calibrating Palmer Drought Severity Index (SC-PDSI) and the Standardized Precipitation Evapotranspiration Index (SPEI) calculated from the climatic data in Beijing, China, both showed major dry periods in 1980-1985 and 1999-2011 (Zhao et al., 2017). Secondly, in term of streamflow, streamflow of the headwater of Baiyang Lake significantly decreased, with tipping point in around 2000 (Fig. S7; Cui et al., 2019; Xu et al., 2019), although precipitation does not show significant trend (Fig. S8). Thirdly according to wavelet analysis, a dampening process in the magnitude of fluctuations in the wavelet coefficients was identified for P and Q after 2000. All these information indicated that the hydrological system has transitioned from wet phase to dry phase around 2000. The wet phase started in 1984 and ended in 2000, while the dry phase was from 2001 to 2015.

Figure 2 showed the comparison of the four signals (P, AET, Q, and ΔS) in the four catchments of the study area. The P signal in the four catchments regularly fluctuated in the wet phase, while dampening fluctuations appeared in the dry phase with weaker amplitudes and lower frequencies. The Q signal was roughly synchronous with the P signal, with similar amplitude and frequency. The ΔS signal showed in-phase connection with P signal in the wet phase, however, the in-phase connection disappeared in the dry phase. The AET signal was out of phase with the other signals in the wet phase, while in the dry phase, AET signal only opposed the ΔS signal in their phases.

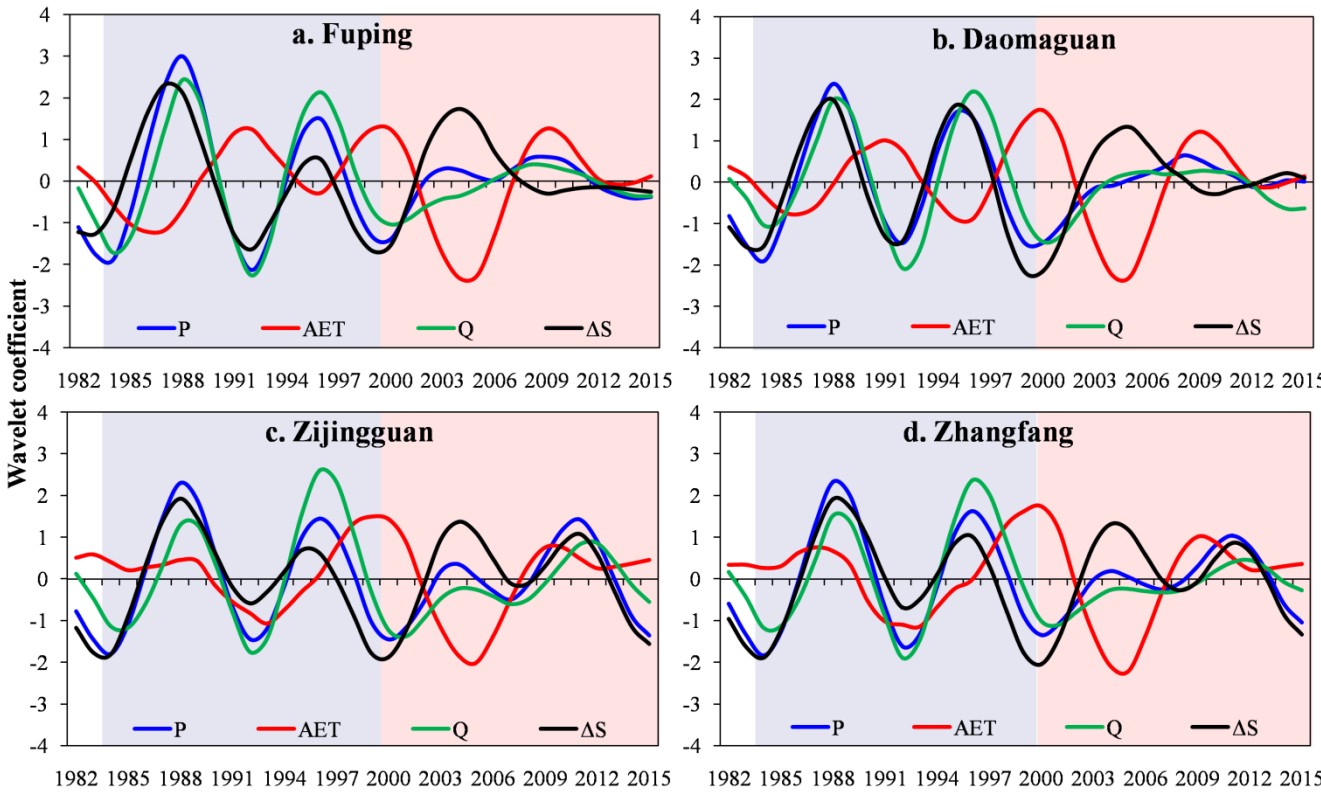

**Figure 2: Wavelet coefficients of four signals (precipitation, P, actual evapotranspiration, AET, streamflow, Q, and catchment water storage change, ΔS) with contraction parameter of 8 in the Fuping catchment (a), Daomaguan catchment (b), Zijingguan catchment (c), and Zhangfang catchment (d). Blue background indicates wet phase from 1984 to 2000, while the red background indicates dry phase from 2001 to 2015.**

The correlation analysis (Fig. 3) corroborated the preliminary interconnection results observed in the wavelet analysis, both for the entire study period and the dry/wet phase period. The AET signal was significantly and negatively correlated with the other signals in the wet phase, but only to the ΔS signal in the dry phase. The Q signal significantly and positively correlated to P signal in both dry and wet phases. The ΔS signal in the wet phase showed significant positive correlations with P and Q signals while negative correlation with AET signal in wet phase. In the dry phase, ΔS signal significantly negatively correlated only to the AET signal. Comparable correlations were observed for all four catchments, with larger and more significant values (e.g., ΔS versus AET) for FP and DMG catchments than ZJG and ZF.

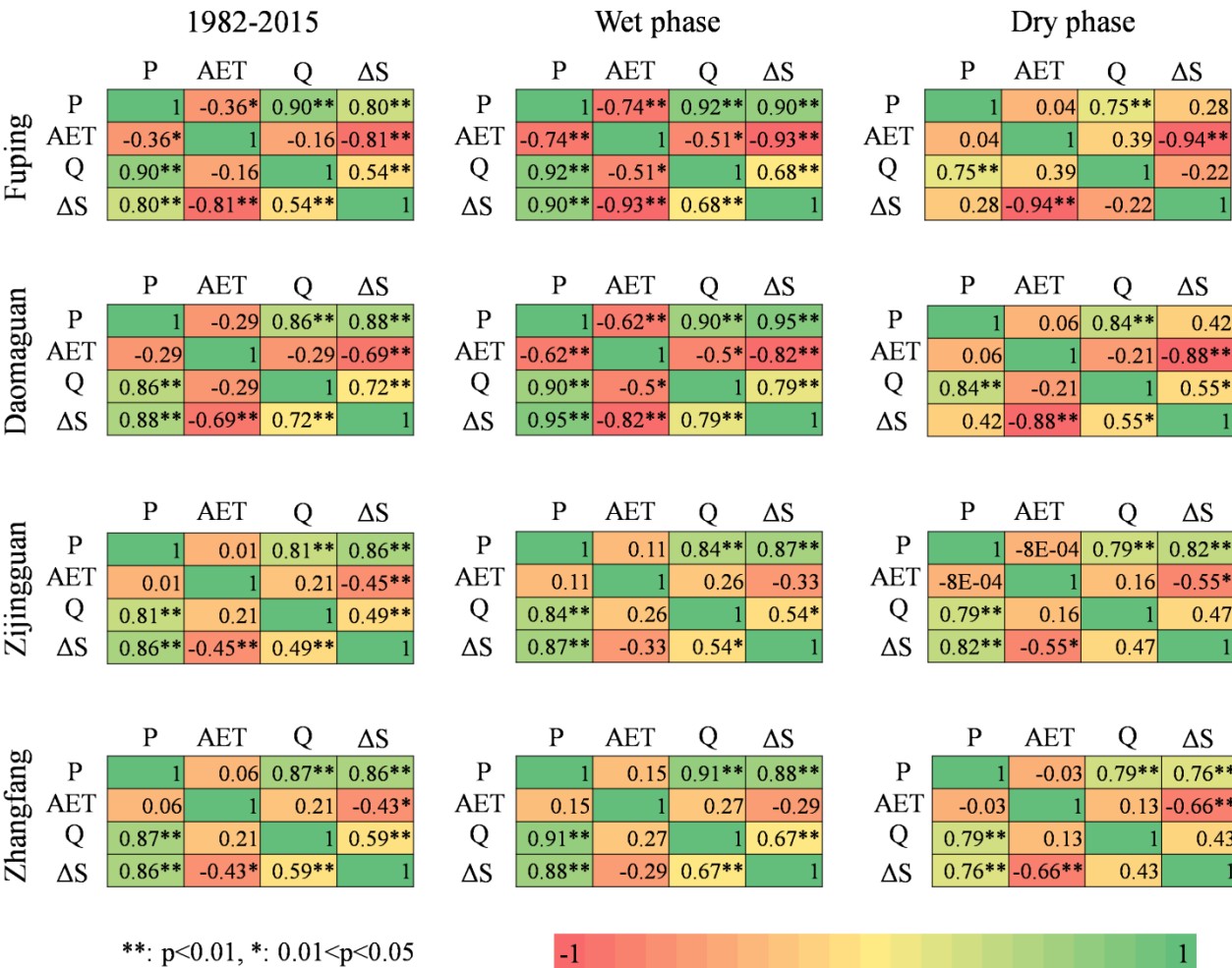

**Figure 3: Pearson's correlation matrices for four variables for the whole study period, the wet phase and the dry phase. The calculation is based on wavelet coefficients with contraction parameter of 8.**

### 3.2 Development of endogenous linking structure of the hydrological system

The causal relationships among variables play an important role in helping to understand the endogenous linking structure of a hydrological system. Taking FP catchment as an example (Fig. 4), the Granger's causality test showed that P does not Granger-cause AET, while it does Granger-cause Q and ΔS; AET does not Granger-cause Q, while it Granger-causes ΔS; ΔS Granger-causes Q and AET. The bidirectional causal relationship between ΔS and AET denotes the existence of unobserved common variables, referred to as confounders (Pearl, 2009; Morgan and Winship, 2015). According to the theory of causal inference, confounders can obscure or blur the "real" causal relationship, thus it is advisable to establish subgroups to analyze the causal relationship separately. In the context of our study, the causal relationship of ΔS and AET is "confounded" by soil water: AET causes ΔS when soil water is plentiful, whereas ΔS causes AET when soil water is scarce.

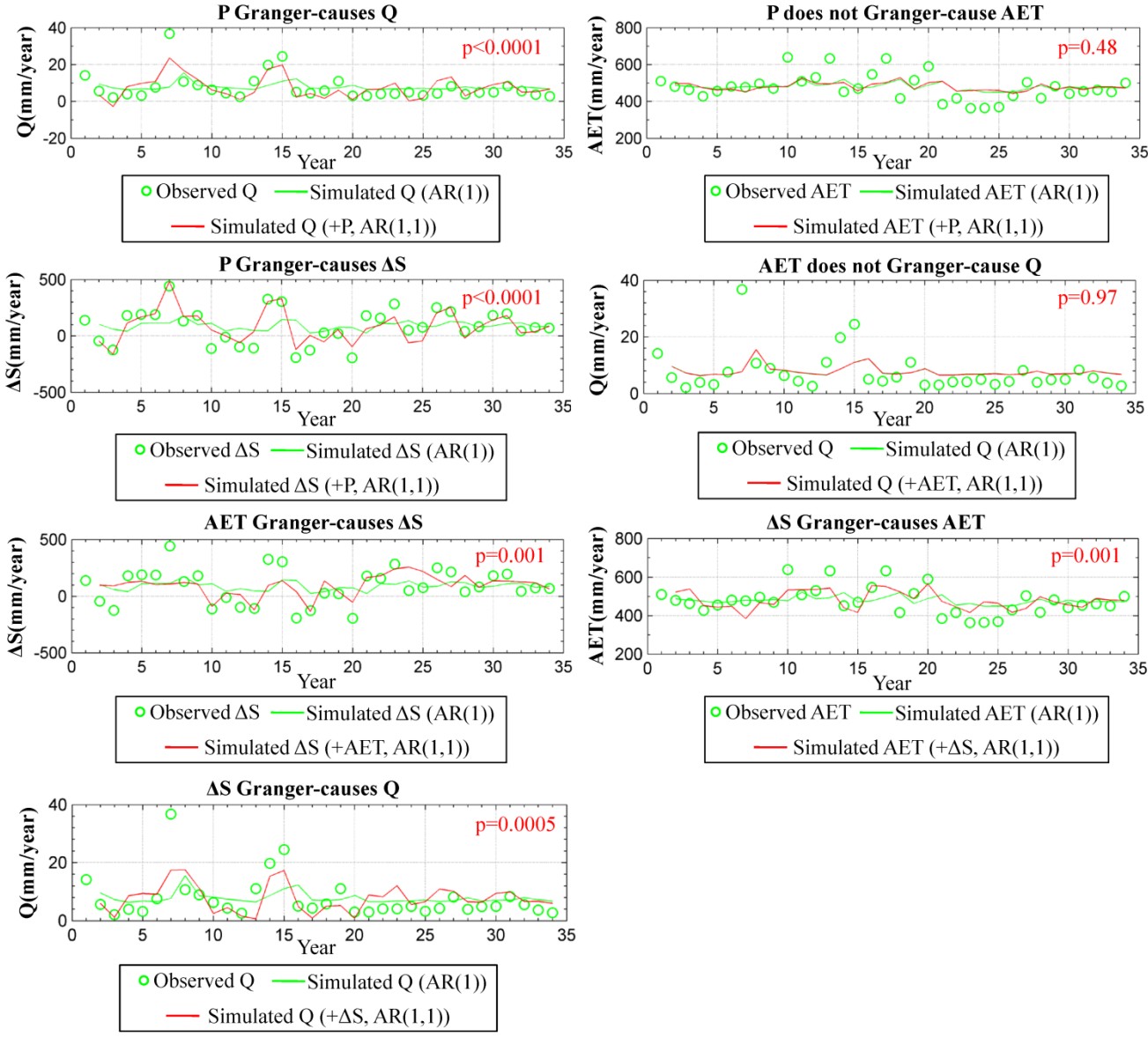

**Figure 4: Causal relationships among four hydrological variables of FP catchment. P, Q, AET, and ΔS indicate rainfall, streamflow, actual evapotranspiration, and catchment water storage change respectively. AR(p) represents the autoregressive model, where "p" is called the order of the model and represents the number of lagged values we want to include. AR(p,q) represents that additional q-order variable is included. The p-value indicates significance at 95% confidence. Comparable results of Granger's causality test were seen for the other catchments (not shown for clarity purpose).**

The endogenous linking structure of the hydrological system based on the Granger's causality test was built in a three-step procedure. First a complete undirected graph is constructed (Fig. 5a), followed by elimination of edges between variables that are independent; in this case the edges between AET and P, and between AET and Q (structure shown in Fig. 5b), and finally providing a direction of the adjacent edges according to the results (Fig. 5c). However, additional knowledge is

necessary to identify the reinforcing/balancing feedback loops and to form the system dynamics' structure of the hydrological system. As shown in Figure 5d, the main trait of the structure built using the system dynamics approach is the soil-vegetation feedback to drive system's dynamic equilibrium, while P was input and Q was influenced by the soil-vegetation feedback structure.

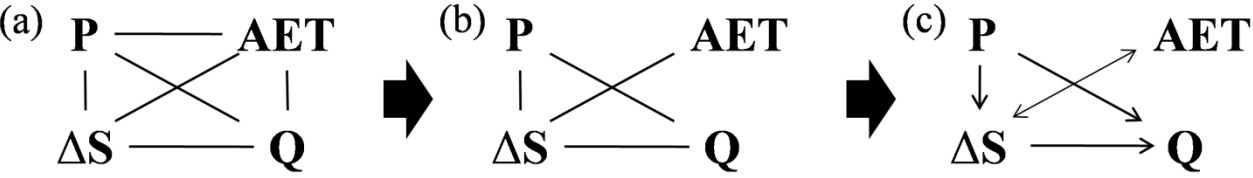

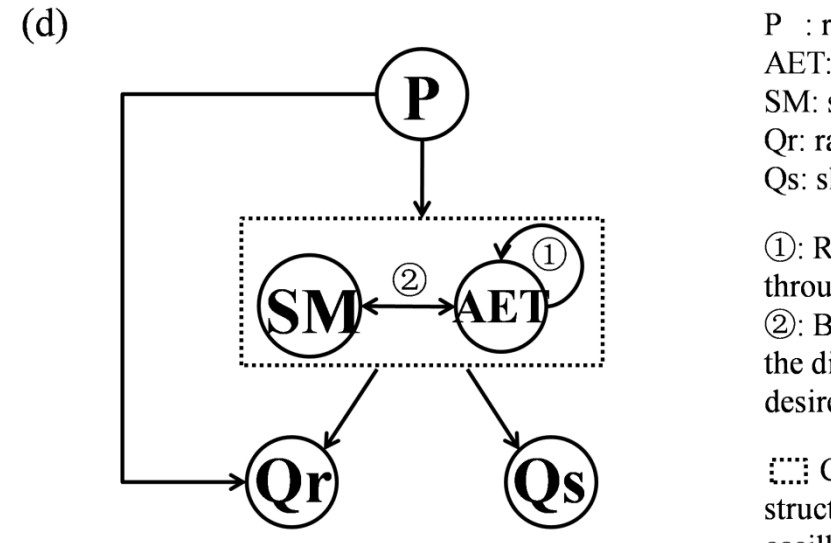

P  : rainfall
AET: actual evapotranspiration
SM: soil moisture
Qr: rapid-response runoff
Qs: slow-response runoff

①: Reinforcing feedback (works through an implicit vegetation stock)
②: Balancing feedback (works through the difference between actual and desired soil moisture)

⌞⌝ Central soil-vegetation feedback structure for producing self-sustained oscillation in long-term

**Figure 5: The comparison of endogenous linking structures of hydrological system built using two approaches, Granger's causality test and system dynamics approach. a-c, illustration of how the structure built using causal analysis, d structure built using system dynamics approach.**

The most interesting knowledge that the system dynamics' structure depicted was that the boundary of the hydrological system should be expanded to incorporate AET (vegetation structure) as an endogenous variable. According to system dynamics theory, exogenous variables should have no feedbacks from or interactions with the stocks inside system boundary (Sterman, 2000). For this study, it was appropriate to exclude P outside the system boundary and consider P as the sole source variable because P was insensitive to AET and SM over small area (Wei and Dirmeyer, 2019). Meanwhile, AET

(vegetation structure) exhibited strong interactions with SM. Therefore, AET (vegetation structure) should be included as an endogenous variable. Over interannual time scale, AET (vegetation structure) was SM-dependent and played an important role in maintaining the dynamic equilibrium of SM. The outflow rate of AET (vegetation structure) could be higher under ample soil water while lower under lack of soil water. Treating AET (vegetation structure) as an exogenous variable may lead to inappropriate depletion rate of SM, as current AET (vegetation structure) was often not related to current climate, which in turn caused further miscalculations of hydrological variables in subsequent periods. Currently, the AET estimation is based on remote-sensing-based vegetation index, which, however, makes the predictions of future AET impossible because the future vegetation index cannot be obtained. Additionally, rapid- and slow-response runoff were considered as sink variables of the hydrological system because their influence on SM and AET could be disregarded once they have coalesced into the riverbed.

### 3.3 Dynamics simulation of the hydrological system

According to the system dynamics equations (3) - (12), using observed P as input and starting with an initial soil moisture state, the dynamics of AET, Q, $\Delta$S (= P - AET- Q) and TWS (total water storage=soil moisture + groundwater + surface water) were simulated for the FP catchment as an example under both natural and human activities scenarios (Fig. 6). Here AET1, Q1, $\Delta$S1 and TWS1 represent the simulated values with constant ESMS, while AET2, Q2, $\Delta$S2 and TWS2 represent the simulated values with varying ESMS. Under natural scenario (GP = 0, VEG = 0), with a constant ESMS value, the simulated AET captured the inter-annual cycles during the study period, despite the lack of annual fluctuations. Simulated Q and $\Delta$S captured both the annual fluctuations and the inter-annual cycles (the *r*-square values of simulated AET, Q, and $\Delta$S are 0.19, 0.21, and 0.82, respectively). However, the discrepancies between observed and simulated AET and $\Delta$S increased over time. If different climatic phases were considered and varying ESMS was adopted to reflect the shift of climatic phase over decadal scale (lower ESMS from 1982-2000 and higher ESMS from 2001-2015), the discrepancies became smaller (the *r*-square values of simulated AET, Q, and $\Delta$S were 0.33, 0.17, and 0.85, respectively).

Evident from Figure 6 was the increase in simulated total water storage after 2000 against an observed anomaly decrease, which was probably caused by human interventions not reflected in the simulation such as groundwater overexploitation and afforestation activity in the study area since 2000, also seen in the increase in the regional NDVI (Fig. S9). Both observed data and literature review were used to set up human activities scenarios. Firstly, given the lack of groundwater pumpage data for the study area and the time frame, we estimated it based on existing literature. The water use of household sector in FP county mainly relies on groundwater and water demand is projected to increase in the future (Wei and Wang, 2019). Furthermore, the groundwater depletion or the decline of water table in North China Plain also showed an exponential increase during the study period (Gong et al., 2018; Lancia et al., 2022; Huang et al., 2023). Secondly, NDVI data showed a linear upward trend during the majority of the study period (Fig. S9), and literature showed that the Taihang Mountain Afforestation Project launched in 1987 and the coverage of vegetation increased 11.2% from 1994 to 2013 (Hu et al., 2017, 2019; Yuan et al., 2019). Therefore two human-intervention scenarios were assumed to linearly increase AET and

exponentially increase groundwater pumpage. When groundwater pumpage scenario was introduced, the simulated AET, ΔS, Q and TWS all greatly improved. When the afforestation scenario was added, it pushed further all the simulated values closer to the observations (the *r*-square values of simulated AET, Q, and ΔS increased notably to 0.35, 0.30, and 0.86, respectively).

     Simulations were conducted for other three catchments with same initial values, parameters, and scenarios (Fig. S10-S12).

Despite small discrepancies caused by the parameterization, for instance, the overestimated Q in ZF catchment, the inter-annual cycles were well captured in all catchments, indicating the robustness of the system dynamic approach. It is thus important to note that system dynamics approach was insensitive to initial values of variables. In addition, the divergence between observed TWSA and simulated TWS in the first several years was probably caused by the inaccurate reconstruction of TWSA series because TWS was less likely decreasing under humid climate and marginal groundwater pumpage.

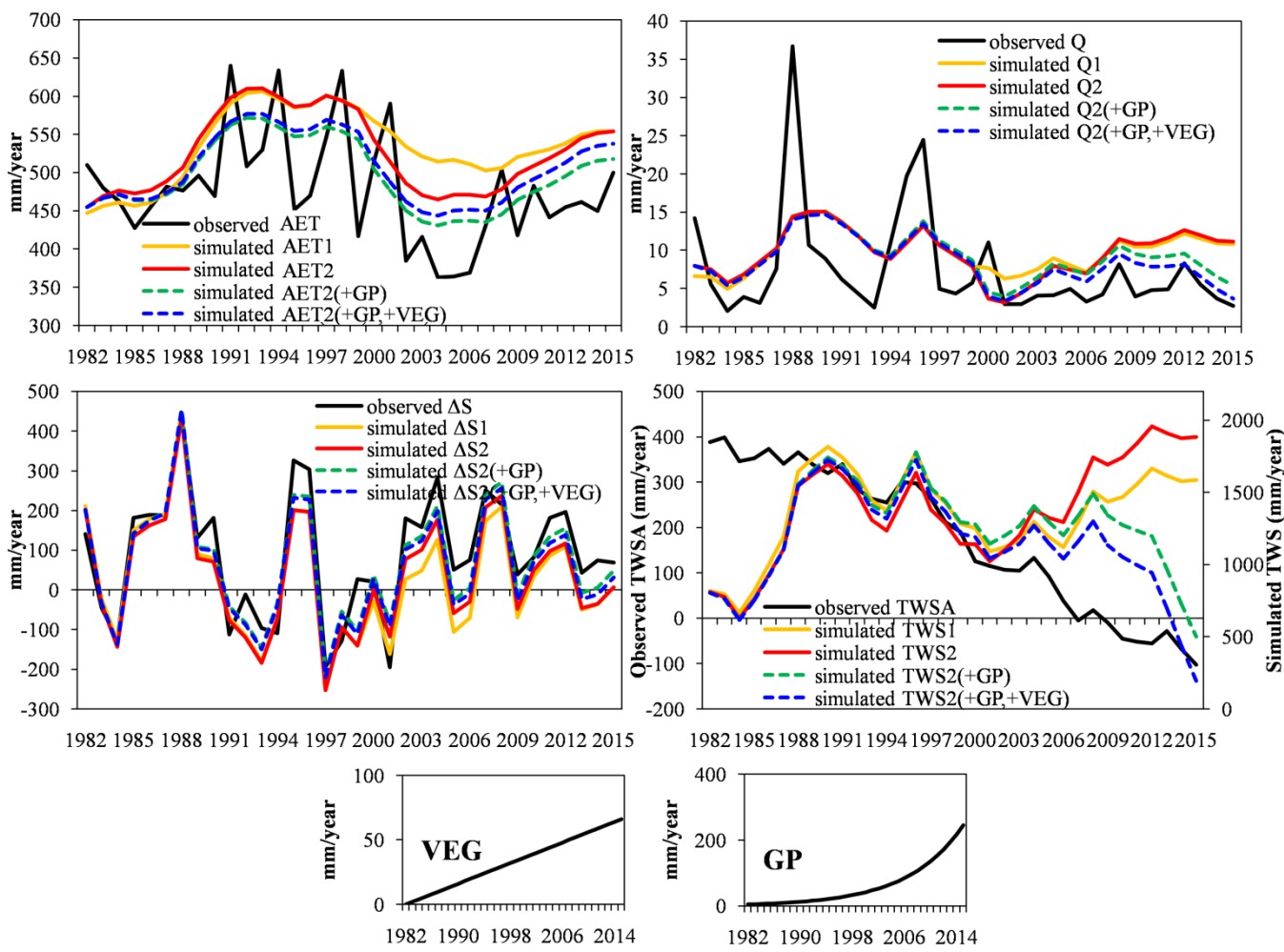

**Figure 6: Observed and simulated actual evapotranspiration (AET), streamflow (Q), catchment water storage change (ΔS) and total water storage (TWS) using system dynamics approach for the FP catchment. Yellow and red lines indicate the simulation**

**without human intervention (GP=0 and VEG=0), with respectively fixed and varying desired soil moisture (ESMS). Green dashed line indicates simulation with groundwater pumpage (GP), which increased at an exponential rate. Blue dash line is the simulation with GP and vegetation change (VEG) which is increasing at a linear rate. AET1, Q1, ΔS1 and TWS1 represent the simulated values with constant ESMS, while AET2, Q2, ΔS2 and TWS2 represent the simulated values with varying ESMS. TWSA is the anomaly of total water storage.**

Using the system dynamics' structure, we computed future runoff and AET from 2015 to 2100 under different SSP+RCP scenarios and compared the results with those from four GCMs. In term of AET, the system dynamics' structure aptly captured its primary behavioral patterns: a decline in 2040s and 2070s, followed by an increase in 2060s and 2090s. Notably, the $r$-squared values between system dynamics' results and GCMs' results were relatively higher for SSP585, specifically 0.48 for ACCESS-CM2, 0.15 for CNRM-ESM2, 0.41 for EC-Earth2, but 0.04 for GFDL-CM4. For the other two SSPs, the $r$-squared values were relatively poor. Regarding runoff, the system dynamics' simulations indicated an increasing trend, whereas the GCMs' estimation did not exhibit any discernible trend. The $r$-squared values between system dynamics' results and GCMs' results for runoff were poor across all experiments. These findings further emphasized the system dynamics' structure is capable of producing long-term hydrological behaviors. Conversely, while adept at capturing short-term fluctuations, process-based models fall short in simulating long-term hydrological behaviors due to lack of the structure.

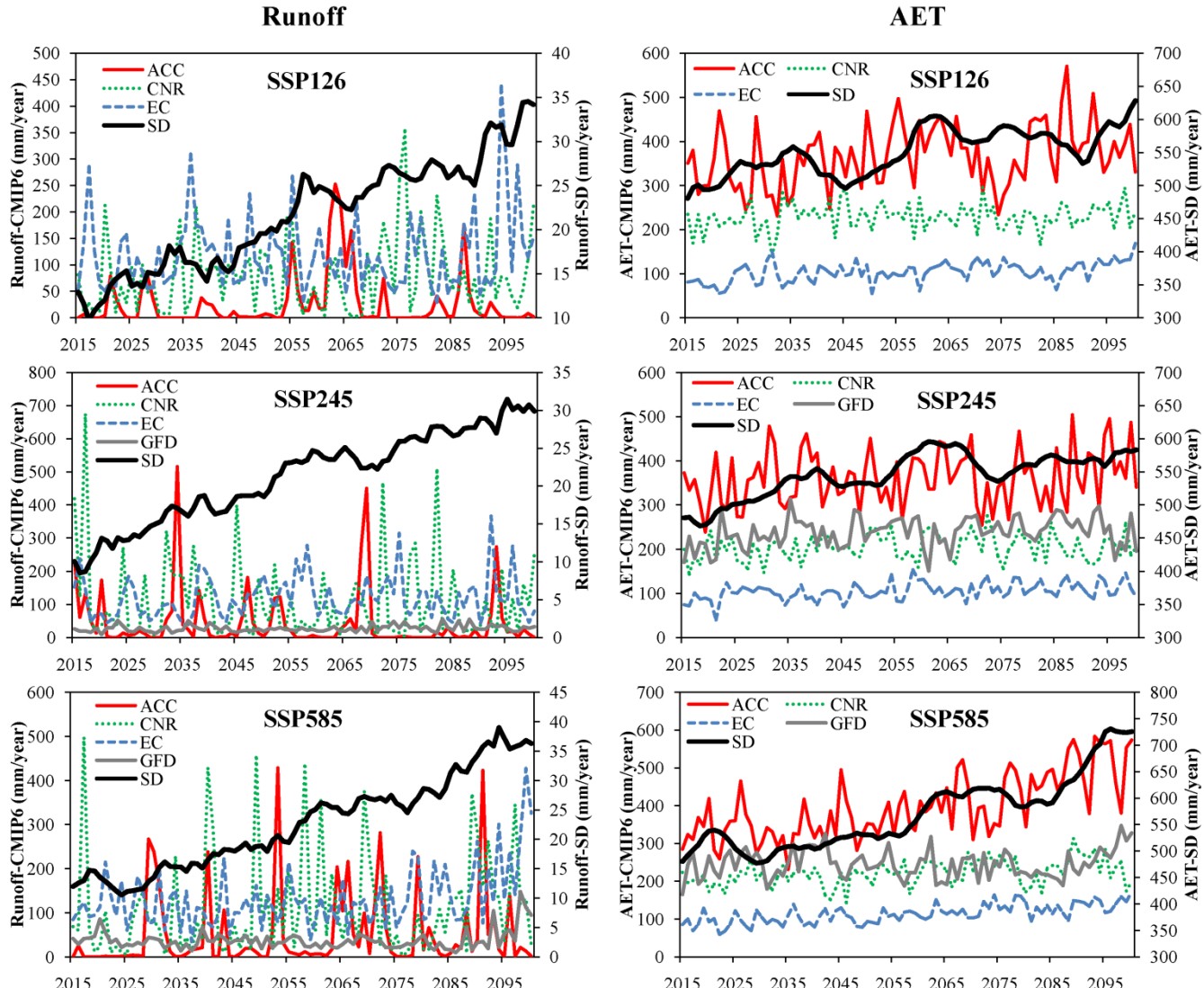

**Figure 7: Future runoff and AET predicted by system dynamics approach under three climatic scenarios and their comparison with predictions from four GCMs. All the dynamics were simulated with fixed ESMS and without human interventions (GP and VEG = 0).**

## 4 Discussions

### 4.1 Distinct mechanisms at different hierarchies of a hydrological system

Here we discuss the interpretation of the underlying mechanisms in term of the hierarchies we defined in hydrological system. At the lowest hierarchy, hydrological processes are primarily controlled by the interaction between rainfall density and sink-filling/macropore flow (McDonnell et al., 2021), which was conceptualized as a small, active reservoir (Xiao et al.,

2019). At this level, soil properties, particularly non-capillary pores in the top layer, are predominant and direct factor influencing the storage-discharge relationship at intra-annual scale. At the higher hierarchy, hydrological processes primarily involve the interaction between vegetation structure change and soil bound water, defined as water stored in capillary pores (Good et al., 2015), which was conceptualized as a large, passive reservoir (Xiao et al., 2019). At this level, soil capillary pores in the deep layer become predominant and direct influencing factor. Water stored in capillary pores is primarily utilized by vegetation, further strengthening the link between vegetation and hydrological behaviors at the inter-annual scale. As ascending to even higher hierarchy, the interaction between climatic oscillation and soil water holding capacity takes over the priority of a hydrological system to generate inter-decadal-scale hydrological dynamics. This is because, soil water holding capacity can be enhanced by the interplays of soil physical, chemical and biological components during drought (Delgado and Gómez, 2016). Physically, drought alters aggregate size fraction with less macro- and more micro-aggregates (Zhang et al., 2019; Su et al., 2020). Also pore network connectivity decreases due to reduced vegetation growth (Nagassa et al., 2015), which likely increases soil water holding capacity under drought conditions because of poorly connected soil pore network and larger water holding capacity of micropores (Patel et al., 2021). Chemically, the ionic strength of soil solution can be up to nine orders of magnitude greater in dry soils compared to saturated soils, creating extremely different chemical environment (Patel et al., 2021). This can cause desorption of different carbon molecules from minerals, and soil organic carbon (SOM) destabilization (Bailey et al., 2019). Biologically, drought stress reduces population and activity of soil animals and microbes. This can also destabilize SOM because the microbial community would preferentially consume the more labile/biodegradable organic molecules, leaving behind a more uniform, stable SOM pool (Kaiser et al., 2015). Furthermore, rainfall pulses in drought period would drive water going deeper because rainfall in dry spells usually occurs as discrete events (Manzoni et al., 2020). These make water easier to be trapped by soil particles rather than utilized by plants, further decreasing evapotranspiration.

The lack of mechanisms in higher hierarchies in conventional hydrological models leads to their hierarchy at intra-annual scale and consequently the failure in producing long-term hydrological dynamics. Firstly, vegetation structure change rather than physiological processes has not been considered. Conventional hydrological models generally employ physiological process-based models to calculate AET. These are derived from experimental data obtained from individual leaf, and then upscaled to canopy by replacing the leaf-scale resistances with their assumed canopy-scale counterparts (Schymanski and Or, 2017). Thus such models still operate on a short-term scale, in which AET is sensitive to environmental turbulent diffusion (characterized by weather data) and vegetation physiological processes (e.g., leaf expansion and abscission) (Chen et al., 2022). However, alongside the spatial scaling up, vegetation structure change, such as growth and morality, takes over the priority to influence AET. This process often takes several years but is ignored in conventional hydrological models. Secondly, the change of soil water holding capacity has not been included. Conventional hydrological models typically assume unvarying soil characteristics and thus stable soil water holding capacity. As the soil structural characteristics change slowly, it is permissible to disregard such changes during simulation when the time span is less than decades. However, these

440 changes become discernible at an inter-decadal scale, thereby necessitating the consideration of alterations in soil structural characteristics.

## 4.2 Advantages and limitations of the system dynamics approach

Firstly, the system dynamics' model in our study is a simplified representation of the complex reality. We argue that both scientific and developmental merits are involved in the proposed model. In essence, the model is a "toy" model. Toy models

are frequently used in theoretical physics and many advanced research scopes (Georgescu, 2012). This is because complex systems usually involve a vast number of interacting elements; however, it is often the case that a small number of factors are quantitatively more significant than the others. Therefore, our simplified model only includes key factors providing a good starting point for building a more complex model (Luczak, 2017). Toy model can help us understand a complex system in its broadest strokes. By breaking it down and then building it back up, we gain profound insights into the system's essence.

By adopting the combined structure of a vegetation reinforcing feedback and a soil water-vegetation balancing feedback, our model significantly outperforms traditional rainfall-runoff models and large models at the long-term scale, while demonstrating marginally inferior performance compared to them at the short-term scale. On one hand, a comparison between our model and SIMHYD, a widely used rainfall-runoff model in Australia (Chiew et al., 2002), was conducted in Fuping catchment as an example. Monthly rainfall and potential evapotranspiration were used as inputs to SIMHYD model,

and monthly results were aggregated to annual values. Then empirical model decomposition method was applied using Matlab to quantify the performance of both models on short-term and long-term scales (Fig. S13). Results showed that SIMHYD model only surpassed our model in short-term Q simulation, but performed poorly in short-term $\Delta S$ simulation and long-term AET, Q, and $\Delta S$ simulation. On the other hand, previous studies have evaluated runoff simulation from global climate (GCMs), global hydrological (GHMs) and land surface models (LSMs). Results showed that all models performed

well in mean annual Q simulation but struggled with the simulation of inter-annual Q change, with median correlation coefficients ($r$) close to 0 for GCMs, and around 0.6 for GHMs and LSMs (r-square values ranging from 0.5-0.6) (Zhou et al., 2012; Hou et al., 2023). Given the study context, our model, with $r$-square values of 0.23 for Q and 0.7 for other variables at the long-term scale (Fig. S13), exhibits comparable performance to the large models, while requiring significantly less data and computational time.

In the future, our model can be improved from at least two aspects. On one hand, the mechanism of low hierarchy in the hydrological system can be integrated into the endogenous linking structure to enhance the model's performance at the short-term scale. As discussed earlier, the interaction between rainfall density and sink-filling/macropores controls the hydrological behaviors at intra-annual scale. Thus, high-temporal-resolution rainfall data and detailed soil properties of the topsoil layer are needed to determine the initiation and duration of short-term runoff events. In addition, to link the two

hierarchies, intra- and inter-annual together, soil macropores/sinks should be modelled as dynamic features that evolve in response to vegetation changes. Previous studies showed that the change of plant productivity affects the input of plant products (above- and belowground litter), causing changes in fractions of particulate organic carbon (Shi et al., 2024),

subsequently affecting soil water-holding capacity and storage-discharge relationship. Furthermore, as topsoil texture becomes finer or coarser, water infiltrating into the deep soil changes, which in turn affects vegetation physiology and structure across scales (Wankmüller et al., 2024). On the other hand, the parameter values should be allowed to vary over time rather than being fixed to improve model's performance at the long-term scale. For instance, the fixed utilization ratio of soil water (C2 in Eq. 8) implies that AET solely depends on vegetation coverage and soil bound water, but not on plant species. However, the dominant plant species in the study area have shifted from herbs to shrubs due to the implementation of afforestation projects (Liu et al., 2011), resulting in significant changes in the utilization ratio of soil water (Liu et al., 2014). By incorporating variable parameters, the model's performance can be enhanced, as the input of high-quality vegetation data, including vegetation coverage and plant species over time, is actually equivalent to the input of AET based on the intrinsically close links between the two factors.

Secondly, ESMS (Eq. 6) as an important parameter represents the soil water holding capacity with value ranging from field capacity to wilting point. Thus, accurate estimation of ESMS is crucial for gaining insights into the long-term hydrological behaviors. However, ESMS is difficult to scale-up from field measurements as below-ground characteristics, such as soil depth and soil porosity, exhibit high heterogeneity and are not readily observable across a catchment. Moreover, field measurements are often labour-intensive, constrained by space and depth limitations. Thus ESMS estimation remains a significant challenge in hydrology. In our study area, soil characteristics are sparsely observed (Fu et al., 2021), and the results obtained from limited soil samples have a strong bias against the plausible distribution. Furthermore, soil structural characteristics undergo gradual changes over time due to the intricate interplay of soil physical, chemical and biological components. Often, field measurement scales are significantly smaller than the scales relevant to changes in soil structural characteristics.

To address these challenges, this study introduced a "top-down" methodology to estimate ESMS, yet, grounded in the system dynamics principles. Following the exogenous interventions, a dynamical system evolves over time, with its trajectories repeatedly passing through a fixed point or multiple points called "attractor" (Rickles et al., 2007). In this context, ESMS can be considered as "attractor" of the hydrological system, which can be inferred from the trajectories of hydrological system dynamics. It is noteworthy that the ESMS is changeable in our study, as the factors that determine ESMS have evolved in tandem with the climate change, corresponding to multiple attractors. Previous studies have also suggested that vegetation metrices, such Gross Primary Productivity (GPP), Leaf Area Index (LAI), can serve as surrogates for soil structure modifications and soil hydraulic properties (Jha et al., 2023). Although vegetation is still considered as exogenous driver in that framework, they highlight the intricate interactions among vegetation, soil structure, and soil moisture.

Finally, the model was developed for a study area characterized by a semi-arid climate with limited water resources. In such regions, energy (temperature and light) is typically abundant and does not constrain vegetation growth. Despite the higher evaporative demand resulting from hotter and drier atmospheric conditions, lack of water hinders the increase in AET by drying out the soil surface, weakening the vegetation, or even causing vegetation death to decrease AET. Therefore, the energy factors such as temperature and light are omitted in our model as the model aims to capture the most critical

mechanism in the study area. The success of the model in replicating nonlinear hydrological dynamics proves that the core factors has been identified. However, for regions with energy limitation, such as the high latitudes and the tropics, energy factors could serve as the primary factors and the soil water-vegetation interaction should be modified into the interaction between energy (temperature/light) and vegetation.

## 5. Conclusions

Using wavelet analysis, Granger's causality test, and ultimately the system dynamics approach for the headwater of Baiyang Lake in China as an example, this study proposed a new modelling approach to describe the mechanism of slow variation of a hydrological system at inter-annual to inter-decadal scales based on four observed variables (P, AET, Q and ΔS) during the period of 1982-2015. Holistic and insightful understanding was achieved on how a hydrological system functions slowly, as summarized below:

(1) Correlation analysis in term of wavelet coefficients showed that there were constantly negative correlations between AET and the ΔS regardless of climatic periodicity. Meanwhile, the correlations between the two factors and other variables differed in wet and dry phases, for instance, Q significantly and positively correlated to the ΔS in the wet phase and lost the link in the dry phase. This implies that interconnections between the AET and the ΔS were robust while the relationships between the two factors and other variables were changing over time.

(2) Causal analysis showed a bidirectional causal relationship between AET and ΔS, suggesting the existence of confounding factor and soil water serves as a "confounder" in this relationship. Thus the causal relationship was subgrouped according to soil water: AET played a dominant role in driving ΔS under plentiful soil water, whereas ΔS determined AET under lack of soil water.

(3) System dynamics approach built an endogenous linking structure among stocks, which comprised a reinforcing feedback representing the vegetation structure change, and a balancing feedback representing the soil water-vegetation relationship. The structure successfully captured the slow behaviors of a hydrological system at inter-annual to inter-decadal scales under both natural and human-intervention scenarios. Our results showed that at an inter-annual scale, the interaction between the vegetation structure change and the soil bound water dominated the hydrological processes, while at an inter-decadal scale, the interaction between the climatic oscillation and soil water holding capacity controlled the hydrological processes.

In conclusion, the system dynamics approach provides a hierarchical view to understand endogenous linking structure of a hydrological system, and has the potential to better reproduce the slow hydrological processes at inter-annual to inter-decadal scales compared to conventional hydrological models. With these insights on the hydrological system, we could advance our knowledge in hydrological science in the century of complexity.

**Financial support**

This work was supported by two grants of the National Natural Science Foundation of China (Grant no. 42301036 and 42171046), and the Open Project Program for Hebei Province Collaborative Innovation Center for Sustainable Utilization of Water Resources and Optimization of Industrial Structure (No. XTZX202101).

**Data Availability**

The gridded data are available in the main text and the catchment-scale data can be obtained by contacting Xinyao Zhou (zhouxy@sjziam.ac.cn) and Yonghui Yang (yonghui.yang@sjziam.ac.cn).

**Author contributions**

Xinyao Zhou: Conceptualization, Methodology, Visualization, Writing — Original draft preparation. Zhuping Sheng and
Kiril Manevski: Methodology, Supervision, Writing — Review & Editing. Rongtian Zhao, Qingzhou Zhang, Yanming Yang, Shumin Han, and Jinghong Liu: Resources. Yonghui Yang: Supervision, Writing — Review & Editing.

**Competing interests**

The contact author has declared that none of the authors has any competing interests.

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
