# Peer review of "Can system dynamics explain long-term hydrological behaviors? The role of endogenous linking structure"

_Hydrology and Earth System Sciences, 2024_

## Author Comment (AC1)

**RC1**: 'Comment on hess-2024-7', Anonymous Referee #1, 02 Apr 2024
**Review for "System dynamics perspective: lack of long-term endogenous feedback accounts for failure of bucket models to replicate slow hydrological behaviors"**

This paper investigates the factors long-term endogenous feedback soil water-vegetation feedback loop through the system dynamics perspective. It is an interesting topic and is widely concerned. However, there are several problems that needs to be revised.

The products of actual evapotranspiration contain much uncertainties. The authors should be very careful with the products and validation of the accuracy in the region is recommended.

Reply: Thank you for the comment. Due to the absence of ET observations during study period, we resorted to three alternative ET products - NTSG (Numerical Terra-dynamic Simulation Group), CR (Complementary-Relationship-based modeling), and PEW (Proportionality hypothesis-based surface Energy-Water balance model) - for cross-validation with ETWatch (as depicted in the figure below). These products employ distinct algorithms, namely Penman-Monteith for NTSG (Zhang et al.., 2009a,b), Complementary-relationship for CR (Ma et al., 2019a,b), and Priestley-Taylor for PEW (Fu et al., 2022a,b). Despite their algorithmic difference, they exhibit common characteristics in long-term data.

Firstly, all the products successfully capture the annual ET fluctuations. Secondly, in terms of long-term trends, all products indicate a decline in ET in the end of 1990s and the beginning of 2000s. While the ET of NTSG and ETWatch shows an upward trend thereafter, however, the ET of CR and PEW continues to decrease in the study period. Thirdly, the magnitude of ET fluctuations is comparable among all the four products, being more pronounced during the wet phase and less so during the dry phase.

Given these observations, we conclude that the ETWatch product demonstrates reliable values and the most distinguishable fluctuations among the four ET products, thus standing out as a reliable choice for ET estimation.

The above paragraph has been added in Method section.

[Figure]

Ma, N., Jozsef, S., Zhang, Y., Liu, W. (2019). Terrestrial evapotranspiration dataset across China (1982-2017). National Tibetan Plateau / Third Pole Environment Data Center. https://doi.org/10.11888/AtmosPhys.tpe.249493.file.

Ma, N., Szilagyi, J., Zhang, Y.S., &Liu, W.B. (2019). Complementary-relationship-based modeling of terrestrial evapotranspiration across China during 1982-2012: Validations and spatiotemporal analyses. Journal of Geophysical Research: Atmospheres, 124.

Fu, J., Wang, W. (2022). Global PEW Land Evapotranspiration Data Set (1982-2018). National Tibetan Plateau / Third Pole Environment Data Center. https://doi.org/10.11888/Terre.tpdc.272874.

Fu, J., Wang, W., Shao, Q., Xing, W., Cao, M., Wei, J., Chen Z., & Nie, W. (2022). Improved global evapotranspiration estimates using proportionality hypothesis-based water balance constraints. Remote Sensing Environ. 279, 113140.

There is a hydrological model adopted. How the parameters are determined, including VEG and GP? Why the model is adopted? To my knowledge, process-based hydrological model is rarely used at annual scale.

Reply: Thank you for the comment. Firstly, the vegetation parameter can be approximated by referencing relevant literature (Hu et al., 2019; Yuan et al., 2019) and analyzing NDVI data. Both sources indicate a linear upward trend during the majority of the study period, as depicted in the figure below. However, given the inaccessibility of groundwater pumpage data for the study area and time frame, we have resorted to estimate it based on existing literature. In the North China Plain,

previous studies have observed an exponential increase in the rate of groundwater depletion or the decline in water table height during the study period (Gong et al., 2018; Lancia et al., 2022; Huang et al., 2023). These approximations have successfully enhanced the model's overall performance.

These explanations will be added in method section and supplementary file.

[Figure]

Second, different from conventional hydrological models, our model is a system dynamics model which primarily emphasizes the interplay between system components. In system dynamics models, the choice of time step is dictated by how the state variables update over time according to your purpose, rather than the infinitesimal time intervals used in traditional numerical models (Naugle et al., 2024). Selecting an appropriate time step is crucial for yielding an acceptable approximation of system dynamics for a given purpose.

This study aims to capture and elucidate hydrological dynamics at a long-term scales. A previous study showed that a clayey-soil catchment tend to exhibit higher flow in the short term but less discharge in the long term than its sandstone counterpart (Xiao et al., 2019). Here, "short-term" refers to time scale of hours to days, while "long-term" refers to annual time scale. This observation aligns with other studies indicating that the mean travel time of a non-groundwater-dominant catchment is around one year or slightly longer (Sterte et al., 2021).

These findings prove that catchment soil water stock typically updates at an annual step, and highlight the distinct mechanisms underlying the short-term and long-term hydrological behaviors. Short-term hydrological processes primarily involve the interaction between rainfall density and sink-filling/macropore flow (McDonnell et al., 2021), making soil properties, particularly the non-capillary pores in the top layer, the predominant and direct influencing factor. Conversely, long-term hydrological

processes are predominantly determined by the difference between current soil water content and soil water holding capacity, thus emphasizing the significance of soil capillary pores in the deep layer. Water stored in capillary pores is defined as "bound water" and is primarily utilized by vegetation (Good et al., 2015), further strengthening the link between vegetation and long-term hydrological behaviors. As the soil structural characteristics change slowly, it is permissible to disregard such changes in short-term simulation. However, these changes become discernible in the long term, thereby necessitating the consideration of alterations in soil structural characteristics.

Please also see the reply to reviewer 2. The above statements have been added in Discussion section.

Huang, Z., Yuan, X., Sun, S., Leng, G., & Tang, Q. (2023). Groundwater depletion rate over China during 1965–2016: The long-term trend and inter-annual variation. Journal of Geophysical Research: Atmospheres, 128, e2022JD038109.

Lancia, M., Yao, Y., Andrews, C.B., Wang, W., Kuang, X., Ni, J., Gorelick, S.M., Scanlon, B.R., Wang, Y., and Zheng, C., 2022. The China groundwater crisis: A mechanistic analysis with implications for global sustainability. Sustainable Horizons, 4, 100042.

Gong, H., Pan, Y., Zheng, L., Li, X., Zhu, L., Zhang, C., Huang, Z., Li, Z., Wang, H., and Zhou, C., 2018. Long-term groundwater storage changes and land subsidence development in the North China Plain. Hydrogeology Journal, 26, 1417-1427.

Hu, S., Wang, F., Zhan, C., Zhao, R., Mo, X., and Liu, L., 2019. Detecting and attributing vegetation changes in Taihang Mountain, China. Journal of Mountain Science, 16(2), 337-350.

Yuan, Z., Yan, D.H., Xu, J.J., Wang, Y.Q., Yao, L.Q., and Yu, Z.Q., 2019. Effects of the precipitation pattern and vegetation coverage variation on the surface runoff characteristics in the eastern Taihang Mountain. Applied Ecology and Environmental Research, 17(3), 5753-5764.

Naugle, A., Langarudi, S., and Clancy, T., 2024. What is (quantitative) system dynamics modeling? Defining characteristics and the opportunities they create. System Dynamics Review, 40(2):e1762.

Xiao, D., Shi, Y., Brantley, S.L., Forsythe, B., DiBiase, R., Davis, K., and Li, L., 2019. Streamflow generation from catchments of contrasting lithologies: the role of soil properties, topography, and catchment size. Water Resources Research, 55, 9234-9257.

Sterte, E.J., Lidman, F., Lindborg, E., Sjoberg, Y., and Laudon, H., 2021. How catchment characteristics influence hydrological pathways and travel time in a boreal landscape. Hydrology and Earth System Sciences, 25, 2133-2158.

McDonnell, J.J., Spence, C., Karran, D.J., van Meerveld, H.J., and Harman, C.J., 2021. Fill-and Spill: A process description of runoff generation at the scale of the beholder. Water Resources Research, 57(5), e2020WR027514.

Good, S.P., Noone, D., and Bowen, G., 2015. Hydrologic connectivity constrains partitioning of global terrestrial water fluxes. Science, 349(6244), 175-177.

The desired (expected) soil moisture stock ESMS is key parameter in the hydrological model. The meaning of ESMS need to be further explained. The discussion in Section 4.1 seems too general.

Reply: Thank you for the comment. The ESMS represents the soil water holding capacity with value ranging from wilting point to field capacity. An accurate estimation of ESMS is crucial for gaining insights into long-term hydrological behaviors. However, ESMS estimation remains a significant challenge in hydrology, primarily due to the complexity of measuring the total volume of soil capillary pores across an entire catchment. The measurement is difficult using the "bottom-up" method as underground characteristics, such as soil depth and soil porosity, exhibit high heterogeneity and are not readily observable. Moreover, such measurements are often labour-intensive, constrained by space and depth limitations. In our study area, soil characteristics are sparsely observed (Fu et al., 2021), and the results obtained from limited soil samples have a strong bias against the plausible distribution. Furthermore, soil structural characteristics undergo gradual changes over time due to the intricate interplay of soil physical, chemical and biological components. Often, the measurement scales are significantly smaller than the scales relevant to changes in soil structural characteristics.

To address these challenges, this study introduces a "top-down" methodology for estimating ESMS, grounded in system dynamics principles. Following the exogenous interventions, a dynamical system evolves over time, with its trajectories repeatedly passing through a fixed point or multiple points called "attractor" (Rickles et al., 2007). In this context, ESMS can be considered as "attractor" of the hydrological system, which can be inferred from the trajectories of hydrological system dynamics. Previous studies have also suggested that vegetation metrices, such Gross Primary Productivity (GPP), Leaf Area Index (LAI), can serve as surrogates for soil structure modifications and soil hydraulic properties (Jha et al., 2023). Although vegetation is still considered as exogenous driver in that framework, they highlight the intricate interactions among vegetation, soil structure, and soil moisture.

The above discussion will be integrated into Discussion section.

Fu, T., Gao, H., Liang, H., and Liu, J., 2021. Controlling factors of soil saturated hydraulic conductivity in Taihang Mountain Region, northern China. Geoderma Regional, 26, e00417.

Rickles, D., Hawe, P., and Shiell, A., 2007. A simple guide to chaos and complexity. Journal of Epidemiology and Community Health, 61, 933-937.

Jha, A., Bonetti, S., Smith, A.P., Souza, R., and Calabrese, S., 2023. Linking soil structure, hydraulic properties, and organic carbon dynamics: A holistic framework to study the impact of climate change and land management. Journal of Geophysical Research: Biogeoscience, 128, e2023JG007389.

Fig 4. What's the meaning of model Z. Could the causal relationship be used to generate better hydrological series compared the the process-based hydrological model?

Reply: Thank you for the comment. Model Z is actually the autoregressive model in Granger's Causality Test and the y-axis is the predicted values from the model. We have revised the code-auto-produced label and changed "model Z" to the name of predicted variables.

Future dynamics of hydrological systems under three climatic scenarios are shown in Fig 7. But could not catch what the key points the author wanted to address.

Reply: Thank you for the comment. The figure has been updated to a new version that contrasts our results with CMIP6 results (as illustrated below). It encompasses monthly total runoff ("mrro" in CMIP6) and evapotranspiration ("evspsblsoil" and "evspsblveg" in CMIP6), stemming from three scenarios (SSP126, SSP245, and SSP585) within four Global Climate Models (ACCESS-CM2, CNRM-ESM2, EC-Earth3, and GFDL-CM4). These data were retrieved from the CMIP6 (the phase 6 of the Coupled Model Intercomparison Project) website (https://esgf-node.llnl.gov/search/cmip6/), coving the time span from 2015 to 2100. Notably, the GFDL-CM4 model lacks the SSP126 experiment, and the ACCESS-CM2 model lacks the "evspsblveg" variable.

The spatial resolution varies, with 250 km for ACCESS-CM2 and CNRM-ESM2 models, and 100 km for EC-Earth2 and GFDL-CM4 models. The values pertainning to the study area were extracted from specific grid cells. For ACCESS-CM2 (CNRM-ESM2), the grid value at the 62nd (83rd) row and 104th (93rd) column, corresponding to longitudes of 115.3125° (115.3125°) and latitudes of 39.375° (39.9218°), were used. Similarly, for EC-Earth3 (GFDL-CM4), the grid values at the 164th (92nd) and 165th (93rd) rows, and the 185th (130th) column, corresponding to longitudes of 114.6094° (114.375°) and 115.3125° (115.625°), and latitudes of 39.649° (39.5°), were utilized.

The unit of kg m$^{-2}$ s$^{-1}$ was converted to mm month$^{-1}$ by multiplying it with 86400 seconds and 30 day. Subsequently, annual values were calculated by summing the monthly values. The average precipitation from different SSP scenarios was then employed to drive system dynamics (SD) model, generating runoff and evapotranspiration estimates.

In term of evapotranspiration (ET), SD model aptly captures its primary behavioral patterns: a decline in the 2040s and 2070s, followed by an increase in the 2060s and 2090s. Notably, the R-squared value between SD model and GCMs for ET is relatively higher for the SSP585 experiment, specifically 0.48 for ACCESS-CM2, 0.15 for CNRM-ESM2, 0.41 for EC-Earth2, and 0.04 for GFDL-CM4. However, for other two experiments, the R-squared values are relatively poor.

Regarding runoff, the SD simulations indicate an increasing trend, whereas the GCMs do not exhibit any discernible trend. The R-squared values between SD and GCMs for runoff are poor across all experiments. These findings further emphasize that process-based models, while adept at capturing short-term fluctuations, may be inadequate in simulating long-term hydrological behaviors.

[Figure]

A general description of the hydrological properties of the catchments is needed.

Reply: Thank you for the comment. The vegetation information has been described in the manuscript, soil properties are added to further illustrate the hydrological properties. "Rainfall is the main source of discharge in the study area (Fu et al., 2024). The soil of Taihang Mountains are primarily developed from granite, gneiss, limestone, and sandstone. Cambisols and luvisols constitute the dominant soil types, accounting for 46.86% and 15.46% of the total area, respectively (Fu et al., 2021). These soil types all have high content of sand gradation, approximately 50%, followed by silt. Clay comprises the smallest proportion of the soil content, at approximately 20% (Yang and Cao, 2021)."

Fu, T., Gao, H., Liang, H., and Liu, J., 2021. Controlling factors of soil saturated hydraulic conductivity in Taihang Mountain Region, northern China. Geoderma Regional, 26, e00417.

Fu, T., Liu, J., Gao, H., Qi, F., Wang, F., and Zhang, M., 2024. Surface and subsurface runoff generation processes and their influencing factors on a hillslope in northern China. Science of The Total Environment, 906, 167372.

Yang, H., and Cao, J., 2021. Analysis of basin morphologic characteristics and their influence on the water yield of mountain watersheds upstream of the Xiongan New Area, North China. Water, 13, 2903.

The use of "bucket models" in the title seems inappropriate.

Reply: Thank you for the comment. We have changed the title into "Multiple hierarchies, distinct mechanisms: the system dynamics simulation of hydrological behaviors at multi-year to decadal scales". Please see the first reply to reviewer 2 for the change.

Line 116. Using $\Delta S$ to indicate water budget seems not suitable.

Reply: Thank you for the comment. We will replace all "$\Delta S$" using "water budget".

Lines 208-211. It is inappropriate to put the discussion there.

Reply: Thank you for the comment. This paragraph will be moved to supplementary materials.

Lines 210-214. The brief description of hydrological characteristics of the wet and dry phase is suggested. Why the wet-dry phase be determined by wavelet coefficient instead of annual precipitation or aridity index?

Reply: Thank you for the comment. To distinguish between wet and dry phase, a comprehensive analysis was undertaken, encompassing wavelet analysis, trend analysis of precipitation and runoff (as figures below), and a literature review about

climate and runoff patterns. Results revealed that precipitation did not exhibit a significant trend, whereas runoff showed a significant decreasing trend during the study period, with a tipping point around 2000, aligning with previous studies. The literature review highlighted that the PDO index, which is closely associated with summer rainfall in East China, underwent a shift from positive to negative phase at the end of 1990s. Furthermore, wavelet analysis indicated a reduction in the magnitude of fluctuations in wavelet coefficients for all 4 variables after 2000. Based on this collective evidence, it was determined that the hydrological system has transitioned from wet phase to dry phase in 2000.

This description will be added in Method section and figures will be added in supplementary document.

[Figure]

MK trend analysis for precipitation

[Figure]

MK trend analysis for runoff

Fig 2. Subtitle of the figure needs to be labelled and also for other figures.

Reply: Thank you for the comment. We labelled the figure subtitle.

Fig 5. What's the meaning of the three sub-figures for Fig 5(a).

Reply: Thank you for the comment. The three sub-figures of Fig 5(a) is a step-by-step process to derive causal structure of hydrological system. More explanations will be added in Figure caption.

Lines 327-328. Penman-Monteith model is used for calculating the potential ET instead of actual ET.

Reply: Thank you for the comment. Reviewing the derivation of the Penman-Monteith (PM) equation is instrumental in understanding how it functions. Initially, Penman integrated the surface energy balance equation with the aerodynamic equation to formulate the Penman equation for estimating evaporation (Dolman et al., 2014). Later, Monteith improved on the Penman equation by incorporating a surface resistance term and a more rigorous term for aerodynamic transfer (Allen, 2005). Therefore, the PM equation originally allowed the direct estimation of evaporation from unsaturated surface by utilizing actual thermodynamics parameters. Subsequently, the FAO56 modified PM equation, transitioning from a one-step approach to the two-step $K_c$-$ET_0$ approach, thereby enhancing its global applicability.

Dolman, A.J., Miralles, D.G., and de Jeu, R.A.M., 2014. Fifty years since Monteith's 1965 seminal paper: the emergence of global ecohydrology. Ecohydrology, 7(3), 897-902.

Allen, 2005. Penman-Monteith equation. Encyclopedia of Soils in the Environment, Elsevier, Academic Press, Pages 180-188.

Line 331. What's the specific of instantaneous scale?

Reply: Thank you for the comment. The instantaneous scale refers to the scale shorter than one hour. The explanation will be integrated into the reply to the second comment, and the scale issue will be discussed comprehensively.

**RC2**: 'Comment on hess-2024-7', Anonymous Referee #2, 18 May 2024
The study uses a system dynamics approach aiming to reveal endogenous factors that account for the long-term slow dynamics of a catchment hydrological system. The paper addresses relevant problem within the scope of HESS, the topic is interesting and widely discussed. However, my recommendation is major revision due to several significant issues in the methodology that seem inadequate for addressing the posed research questions. Further justification is also necessary for the conclusions made. Below are my main concerns.

First, I did not find confirmation that the studied basin demonstrates "slow hydrological behaviors." If I understand correctly, the authors believe that such confirmation is provided by the obtained results of wavelet analysis. I do not think so. Wavelet analysis evaluates the coincidence or difference in the phases of oscillations of various hydrological variables, which can be useful for assessing the cause-and-effect relationships between them as well as for identifying long-term wet-dry periods, as was shown in the paper. But the results presented, in particular the fact of a lag in the response of evapotranspiration to precipitation, do not indicate per se the slow behavior of the system. Such evidence could follow from an analysis of the presence of a long-memory effect in the studied hydrological time series. (Note, that in a paper devoted to the problems of detecting slow hydrological behaviors and the physical mechanisms that control this behavior, it would be appropriate to reference at least the most well-known hydrological publications in this area (e.g., some of them cited by O'Connell et al. (2016)). However, the results of the description of hydrological processes in the studied catchments using standard autoregressive models presented in the paper (Fig. 4) allow us to doubt that these are long-memory processes.

Reply: Thank you for the comment and encouragement. As a complex system, hydrological system comprises multiple hierarchies. Each hierarchy is governed by distinct mechanism and produces fluctuations at certain time scale. At the lowest hierarchy of hydrologic systems, runoff generation is primarily influenced by the interaction between rainfall density and macropores/sinks (Uhlenbrook, 2006; McDonnell, 2013), yielding fluctuations at a scale from hours to days. At a higher hierarchy, runoff generation is controlled by the interaction between soil bound water and vegetation growth, leading to yearly to multi-year hydrological cycles. At an even higher hierarchy, runoff generation is determined by the interaction between climate oscillations and soil water holding capacity, resulting in hydrological cycles at a scale of a decade to multi-decades. As we ascend to higher hierarchy, glacial and geological transformations also dictate runoff patterns, introducing hydrological cycles over centuries. Notably, the time series yielded at each hierarchy can be considered as "long-term" or "slow" compared to its predecessor. This study aims to address hydrological dynamics at multi-year to decadal scales. Our results also showed that the multi-year hydrological cycle can be explained by soil water-vegetation interaction, and the change of soil water holding capacity in different climate phase can improve the simulated hydrological dynamics at decadal scale.

The above discussion will be integrated into Discussion section.

Uhlenbrook, S., 2006. Catchment hydrology -- a science in which all processes are preferential. Hydrological Processes, 20, 3581-3585.

McDonnell, J.J., 2013. Are all runoff processes the same? Hydrological Processes, 27, 4103-4111.

Second, if there is evidence that the dynamics of the system can be interpreted in the desired way, then the use of standard wavelet analysis, which was developed for processes with short memory, is questionable. For such processes, the analysis has to be modified (see, e.g., Percival and Guttorp, 1994; Hsu, 2006).

Reply: Thank you for the comment. As previously discussed, this paper aims to elucidate the driving mechanism of hydrological cycles at multi-year to decadal scales. Wavelet analysis is a commonly used tool to identify these varying cycles. We will change the words of "long"/"short" to specific time scales.

Third, as an alternative that allows one, in contrast to the bucket model, to describe "slow hydrological behaviors," a simple linear model of annual changes in the components of the water balance of the river basin under study is proposed. In the paper, I was unable to find results demonstrating that such a model has an advantage in describing the slow dynamics of a system, so I see no reason to consider the proposed model a reasonable alternative. The calculation results for ET, Q and TWS (Fig. 6 and Figs. in Suppl. Materials) are poor, i.e. the suggested model not only does not reproduce the desired effects, but also does not meet the performance measures adopted for hydrological models (the values of the coefficient of determination given in the paper confirm my opinion).

Reply: Thank you for the comment. First, despite its simplicity, our model effectively captures the nonlinear hydrological behaviors occurring at multi-year to decadal scales. In essence, our model is a toy model, also known as minimal model or exploratory model. Toy models are used with pride in theoretical physics and many advanced research scopes (Georgescu, 2012). Complex systems usually involve a vast number of interacting elements, however, it is often the case that a small number of factors are quantitatively more significant than the others. Therefore, looking at a highly simplified model that only includes those key factors can provide an excellent starting point for building a more complex model (Luczak, 2017). We refer to this as a toy model. In fact, finding a simple core hiding in a complex system is often among the most creative and insightful advances in science. Playing with toy model can help us understand a complex system in its broadest strokes. By breaking it down and then building it back up, we gain profound insights into the system's essence.

Second, on one hand, since our model primarily focuses on hydrological behaviors occurring at multi-year to decadal scales, it may not accurately represent mechanisms

operating on shorter time scales. Consequently, instantaneous events such as flash floods may not be captured by our model, potentially affecting its overall performance. On the other hand, our model's performance is not significantly inferior compared to large models. Previous studies have evaluated runoff simulation from global climate (GCMs), global hydrological (GHMs) and land surface models (LSMs). Results showed that the GCMs failed in capturing observed runoff with median value of r close to 0, while GHMs and LSMs can capture observed runoff with median value of r higher than 0.6 (r-square values ranging from 0.5-0.6) (Zhou et al., 2012; Hou et al., 2023). Given this, our model (r-square values around 0.3) exhibits comparable performance to large models, while requiring significantly less data and computational time.

The above discussion will be integrated into Discussion section.

Georgescu, I., 2012. Toy model. Nature Physics, 8, 444.

Luczak, J., 2017. Talk about toy models. Studies in History and Philosophy of Science Part B: Studies in History and Philosophy of Modern Physics, 57, 1-17.

Hou, Y., Guo, H., Yang, Y., and Liu, W., 2023. Global evaluation of runoff simulation from climate, hydrological and land surface models. Water Resources Research, 59, e2021WR031817.

Zhou, X., Zhang, Y., Wang, Y., Zhang, H., Vaze, J., Zhang, L., Yang, Y., and Zhou, Y., 2012. Benchmarking global land surface models against the observed mean annual runoff from 150 large basins. Journal of Hydrology, 470-471(2012), 269-279.

Thus, I have to conclude that the title of the paper does not reflect the content of its current version, since it does not provide any evidence that "lack of long-term endogenous feedback accounts for failure of bucket models to replicate slow hydrological behaviors."

Reply: Thank you for the comment. We have changed our title into "Multiple hierarchies, distinct mechanisms: the system dynamics simulation of hydrological behaviors at multi-year to decadal scales".

Nan-Jung Hsu (2006) Long-memory wavelet models. Statistica Sinica 16, 1255-1271

O'Connell P.E. et al. (2016) The scientific legacy of Harold Edwin Hurst (1880–1978), Hydrological Sciences Journal, 61:9, 1571-1590, DOI: 10.1080/02626667.2015.1125998

Percival, D.B. and Guttorp, P. (1994) Long-Memory Processes, the Allan Variance and Wavelets, Editor(s): Efi Foufoula-Georgiou, Praveen Kumar, Wavelet Analysis and Its Applications, Academic Press, 4, 325-344

**Specific and technical comments**

1. Lines 148-151: This describes the Granger's Causality Test for determining the dependence of X on Y (Y causes or does not cause X), while equation (2) describes the dependence of Y on X.Change either the description on lines 148-151 or equation (2).

   Reply: Thank you for the comment. We have changed the description accordingly.

2. Line 239: If ET Granger-causes S, and $\Delta$S Granger-causes ET, then the question is: what is the cause and what is the effect?

   Reply: Thank you for the comment. According to the theory of causal inference, edges in a causal graph are inherently uni-directed. The presence of a bidirected edge denotes the existence of unobserved common variables, referred to as *confounders* (Pearl, 2009; Morgan and Winship, 2015). Confounders can obscure or blur the "real" causal relationship, thus are regarded as undesirable. In the context of our study, the causal effect of $\Delta$S and ET is "confounded" by soil water. More concisely, soil water serves as a "confounder" in this relationship. To address this, it is advisable to establish subgroups to analyze the causal relationship separately: ET causes $\Delta$S when soil water is abundant, whereas $\Delta$S causes ET when soil water is scarce.

   Pearl, J., 2009. Causality: Models, reasoning, and inference (second edition). Cambridge University Press, New York.

   Morgan, S.L., and Winship, C., 2015. Counterfactuals and causal inference. Methods and Principles for social research (second edition). Cambridge University Press, New York.

3. Fig. 4: Change "cause" to "causes".

   Reply: Thank you for the comment. We have changed it accordingly.

4. Fig. 4: What does AR(1) mean? First order autoregressive model? What, then, is AR(1,1)? Explanation required.

   Reply: Thank you for the comment. AR (1) means the autoregression model with 1 time step difference. We have clarified it in the figure caption.

5. Fig. 5: It is not clear what results allowed the authors to conclude that reinforcing feedback (1) exists.

   Reply: Thank you for the comment. Taking vegetation growth as reinforcing feedback is from collective prior knowledge including population ecology,

resource competition theory, and so on (Craine and Dybzinski, 2013; Snider and Brimlow, 2013; Lian et al., 2021; Wright and Francia, 2024). This has been explained in Method section.

Wright, A.J., Francia, R.M., 2024. Plant traits, microclimate temperature and humidity: A research agenda for advancing nature-based solution to a warming and drying climate. Journal of Ecology, 00:1-9.

Craine, J.M., and Dybzinski, R., 2013. Mechanisms of plant competition for nutrients, water and light. Functional Ecology, 27(4), 833-840.

Snider, S.B., and Brimlow, J.N., 2013. An introduction to population growth. Nature Education Knowledge, 4(4):3.

Lian, X., Piao, S., Chen, A., et al., 2021. Seasonal biological carryover dominates northern vegetation growth. Nature Communications, 12, 983.

6. Line 257: Until now, there has been no talk about hysteresis effect. Explanation required.

   Reply: Thank you for the comment. This sentence has been deleted.

7. Lines 268-270: It is necessary to expand the description of the solution to equations (3)-(12). In particular, explain how the variables VEG(t), K(t), ESMS are determined (in the equations, the latter is designated as a constant). How are the constants C1, C2, C3, C4 determined?

   Reply: Thank you for the comment. The initial values of these parameters are first obtained from literature, then calibrated against observed ET and discharge. Further explanations have been added in Method section.

8. Lines 271-272: The presented results do not confirm the statement that "Simulated Q and $\Delta S$ captured both the annual fluctuations and the long-term trends".

   Reply: Thank you for the comment. We modified the sentence with specific time scales added.

9. Fig. 6 and Figs. in Suppl. Materials: The calculation results for ET, Q and TWS are poor. Herewith, good results were obtained for $\Delta S$. Why? I believe that these results are achieved by adjusting the calibration factors and variables. In this case, coefficients C1-C4 have no physical content and can take on any values. For $\Delta S$, this adjustment made it possible to compensate for the poor calculation results of other variables. Please comment.

Reply: Thank you for the comment. As mentioned above, we calibrated our model against observed ET and discharge but not $\Delta S$. The high accuracy of $\Delta S$ is achieved due to the slow rate of change in soil water, which is compatible with the time scale we are primarily focusing on. Moreover, the coefficients C1-C4 have physical content. For example, C1 is the proportion of impermeable area in a catchment. We have added the physical contents of these parameters at Method section.

10. Fig. 6: What is ET1, ET2, Q1, Q2, $\Delta S1$, $\Delta S2$, TWS1, TWS2? What is TWSA?

Reply: Thank you for the comments. The further explanations has been added in figure caption.

11. Lines 298-302: How were anthropogenic impacts (VEG, GP) set for the future period? How were coefficients C1-C4 set? The same as for the historical period? On what basis, if these are purely empirical coefficients reflecting data for the observation period? Overall, I see no point in using an ineffective hydrological model to estimate the future state of a hydrological system. In addition, these experiments are not relevant to the main content of the paper. I suggest removing them.

Reply: Thank you for the comment. Please refers to the reply to the 9th specific comments. Furthermore, we temporarily set VEG and GP as 0 and fix C1-C4 in the future projection. These parameters may change over time. In this study, however, we only show the simplest model and want to capture the essence of hydrological dynamics at multi-year to decadal scales. In the future work, with more details added in, these parameters can be changeable to get more accurate results.

12. The reasoning in subsection 4.1 is correct in essence but has no relation to the results obtained. The listed physical, chemical, and biological mechanisms influencing changes in soil water retention are not described by the extremely simple model proposed. Therefore, the fact that the desired soil moisture turned out to be higher in the dry phase has nothing to do with these mechanisms but, as I assume, is only an accidental consequence of the calibration procedure.

Reply: Thank you for the comment. Please refers to the reply to the first comment. The slow shift of soil water holding capacity with the change of capillary pores, and its influence on hydrological cycles have been captured by our model due to the improvement of model performance with higher ESMS in the dry phase. The more detailed discussion will be provided in manuscript.

13. Also, the reasoning in subsection 4.2 is not relevant to the results obtained. The previous sections do not show that vegetation changes, "such as tree

growth and mortality, have become significant factors influencing ET over climate". It is not clear what long-term delay in hydrological response the authors are talking about. No results were presented to support the presence of such relationships. The conclusion that "persistent hydrological shifts and especially flow reductions such as those caused by the increasing and enduring multi-year drought can only be described accurately with the system dynamics approach" is not supported by the presented results.

Reply: Thank you for the comment. According to the reply to the first comment, we will reorganize the discussion and focus on the different hierarchies that conventional hydrological models and our model in. We believe our study contributes to the research gap identified by Fowler et al. (2020) on improved understanding of the hydrological dynamics at multi-year to decadal scales.

**RC3**: 'Comment on hess-2024-7', Anonymous Referee #3, 22 May 2024

This study seeks to look at the problem of "slow hydrological behaviours" via a different lens, namely that of system dynamics. I certainly agree that this is a pressing problem that needs some fresh thinking, since many current "bucket" models do a poor job at simulating such behaviours. The system dynamics approach may be relevant here, but I'm concerned that there are multiple serious issues with the implementation here, as follows.

- **Missing processes, and conceptual confusion, regarding evapotranspiration.** The authors don't seem to be taking into account the concept of evaporative demand - that is, they are neglecting the fact that the actual ET is a function of two things: (1) how hot and dry the near-surface atmosphere is, as captured in the concept of Potential Evapotranspiration (PET); and (2) the subsurface water availability, eg. in soil moisture. (1) seems to be missing so let's focus on that. At line 177 it says "while traditional hydrological models usually use physical process-based models such as the Penman or the Thornthwaite models to calculate ET...". [NB: By "ET", I believe the authors here mean actual evapotranspiration (AET), because this is how they have used that acronym throughout the manuscript - which is part of the confusion.] Anyway, no, that is not what "traditional" models do - typically, methods such as Penman are used to estimate PET, not AET; such methods produce a PET timeseries which becomes an *input* to the modelling processes. In other words, whereas the modelling here seems to use only one input, precipitation, traditional modelling uses two, precipitation and PET. So the present method seems to be neglecting the reality that a hotter drier atmosphere can result in a greater proportion of precipitation being lost to AET, all else being equal. I presume the same framework could be altered to add this additional driving variable, but I'm unsure. In any case, the modelling is subsequently applied to a climate change scenario which invariably means a hotter (if not drier) world, and yet the modelling seemingly cannot account for one of the most basic elements of the climate change signal (ie. rising temperatures)—this is unacceptable. I note that I may have misunderstood something here so I'd be happy to be corrected by the authors.

Reply: Thank you for the comment. First, our model is developed specifically for our study area, characterized by a semi-arid climate with limited water resources. In such regions, energy (temperature and light) is typically abundant and does not constrain vegetation growth or actual evapotranspiration (AET). Despite the higher evaporative demand resulting from hotter and drier atmospheric conditions, lack of water will hinder the increase in AET by drying out the soil surface, weakening the vegetation, or even causing vegetation death to decrease AET. Therefore the energy factors such as temperature and light are omitted in our model as our model aims to capture the most critical mechanism in our study area (Please see the reply to reviewer 2). The success of our model in replicating nonlinear hydrological dynamics proves that the core factors has been identified. Nevertheless, for regions with energy limitation, such as high latitudes and tropic regions, energy factors could serve as the primary factors.

Then the soil water-vegetation interaction can be modified into the interaction between temperature/light and vegetation.

Additionally, the Penman-Monteith equation was originally developed to estimate AET when actual thermodynamic parameters are used. Please refer to our response to reviewer 1 for more details.

Second, the subsurface water availability is one of core factors in our model, which implicitly included in our time scales. Please see the replies to reviewer 1 and 2 for detailed description.

We apologize for any lack of clarity in explaining the background and context of our work. The above discussion will be integrated into Discussion section.

- **The chosen approach induces memory that is too long for some system components**.   The calculation of key fluxes (AET, Q, recharge) depends upon the discrepancy, DISC, between soil moisture and the assumed ideal or "expected" soil moisture storage. The trouble is that since the equations are being solved at a yearly timestep, this means the AET, Q and recharge don't respond to a high rainfall value until the year after it happens.   Basic common sense would seem to ward against this - ie. the flood doesn't come the year after the rainstorm.   I wonder if this could be solved if the same approach were used on a shorter time step and/or if the equations were solved in such a way that the equations took account of forcing fluxes from the same timestep than the one being solved.

Reply: Thank you for the comment. Please refer to the first reply to reviewer 2. Hydrological system comprises multiple hierarchies, with each hierarchy characterized by a distinct primary driving mechanism that generates time series fluctuating at different time scales. Our study focus on hydrological dynamics occurring at multi-year to decadal scales. Specifically, the interaction between soil bound water and vegetation growth is dominant at the multi-year scale, whereas the interaction between climate oscillation and soil water holding capacity is dominant at the decadal scale. In addition, our model, albeit the minimal structure in design, serves as a conceptual framework to elucidate the fundamental hydrological dynamics at multi-year to decadal scales. We believe that in the future, as our model evolves to incorporate more intricate details, additional mechanisms and factors will emerge to enhance our understanding of hydrological dynamics across the various hierarchies.

- **The conceptualisation is seemingly ill-suited in the case of streamflow.**  The observed peaks in streamflow are absent, associated with severly underestimated streamflow variability.   The streamflow is calculated as the aforementioned discrepancy in soil moisture multiplied by a calibratable parameter.   Since soil moisture is a state variable that is only allowed to vary on annual timesteps, it is rather slow moving, not episodic, and this property is then translated to streamflow dynamics as well.

Reply: Thank you for the comment. Please see the reply to previous comment and reviewer 2. You precisely capture the key point in our model and we will discuss it in the discussion section.

MINOR COMMENTS:

Line 109: I suggest that "AET" is used instead of "ET" throughout - this will minimise confusion with PET

Reply: Thank you for the comment. All the "ET" has changed to "AET" according to the comment.

117: It would be good to discuss the uncertainty of information being used here, particularly remotely sensed AET.   Given the uncertainty in the RS AET, it's ill advised to calculate change in S by subtracting RS AET from precip (for an example of how a study accounted for this uncertainty by water-balance-based factoring, see here: https://doi.org/10.1029/2022WR033538).

Reply: Thank you for the comment. Please refers to the reply to reviewer 1. We have downloaded three additional AET products and compared them with the ETWatch product. The four AET products are produced using different algorithm. Generally, they are equivalent in values but do show uncertainties in fluctuations and trends. Among them, the ETWatch product stands out for displaying the most prominent fluctuations. This comparison will be added in Method section.

136: Many readers will not have a background in the concepts and/or methods used.   Please explain the concept of a mother wavelet.

Reply: Thank you for the comment. More explanation about mother wavelet has been added in Method section as below:

$\Psi(t)$ is known as mother wavelet because it can generate child wavelets by dilation and translation. Function is then processed with these child wavelets to yield wavelet coefficient (Sayood, 2012).

Sayood, K., 2012. Wavelets. Introduction to Data Compression (Fourth Edition). Elsevier. doi: http://dx.doi.org/10.1016/B978-0-12-415796-5.00015-6.

In general, much of the language used anthropomorphises the system, such as saying it has a "goal" ("[balancing feedback loops are] aiming for stability").   Also, where it is said that it "desires" a certain soil moisture.

Reply: Thank you for the comment. The "desired" is used in system dynamics due to it is often used to simulate economic dynamics. For a balancing feedback loop, its "goal" is to keep system stable by approaching the "desired" value.

196: Better to call it "precipitation input" rather than "precipitation inflow" as the latter is too close to language used for streamflow. Likewise the other "outflows" would be better as "outputs".

Reply: Thank you for the comment. The "inflow" and "outflow" are typically used in system dynamics and correspond to "stock". We think that the nonlinear dynamics can be better implicitly represented by these terms.

200: "Correction coefficients" makes it sound like something is incorrect. Perhaps "coefficients that determine the rapidity with which the system self-corrects after a disturbance" or "responds after a disturbance"

Reply: Thank you for the comment. We have changed the blurred description into clearer one according to this suggestion.

215: This is method and needs to be moved to the methods section.

Reply: Thank you for the comment. This sentence has been moved to method section accordingly.

Figure 2. The choice of colour is confusing - Greens and blues are typically reserved for fluxes like precip and streamflow, while reds and yellows for ET. So, swap colour of P and ET.

Reply: Thank you for the comment. The figure has been revised as this suggestion as below.

[Figure]

252 onwards: There needs to be a subsection in the methods section that explains the broad logic of doing this (even though it won't yet be possible to be specific since the results are yet to be presented). That is, you need to say that you build the model based on the causal links seen in separate analysis.

Reply: Thank you for the comment. Indeed, the wavelet analysis, causal inference, and system dynamics all provide a distinctive perspective for us to understand hydrological system. However, it is important to note that they operate in parallel rather than forming a causal chain. For example, the concept of feedback in system dynamics does not find its counterpart in the other approaches. We will incorporate an additional subsection in method section to elucidate their mutual relationship among these methods.

264: the remotely-sensed vegetation indices are discussed as if the authors are unaware that remotely-sensed AET is itself calculated from one of these indices (eg. NDVI)

Reply: Thank you for the comment. The sentence has been revised into "Remote-sensing-based vegetation index is necessary for AET estimation, however, the prediction for future AET becomes impossible because the future vegetation data cannot be obtained."

301-02: No, that's not the way to interpret GCM sequences because, although it is hoped that their climate sequences are realistic overall (ie. have similar statistics to reality for the sequence pre-2024), the exact timing of dry periods and wet periods is decoupled from reality; ie. the simulated sequence is purely synthetic (thus, the

droughts that appear in the pre-2024 simulated sequence are not timed with historic droughts).

Reply: Thank you for the comment. Please see the reply to reviewer 1. Figure 7 has changed into a comparative analysis of the results between our model and 4 Global Climate Models (GCMs). This comparison reveals both consistencies and inconsistencies. Interestingly, the projected runoff by GCMs does not show long-term trend, while our model predicts a gradual increase in runoff in the future.

Figure 6: the bottom right hand panel is cheating a bit: while it's ok to shift the lines vertically to emphasise the match, the rate of change per unit length should be identical. In other words, since the axis range on the right is 2000 (ie. simulated TWSA has maximum 2000, minimum 0), the other axis (for observed TWSA) should have a range of 2000 also (ie. perhaps -500 to +1500). The current arrangement artificially inflates the apparent match.

Reply: Thank you for the comment. The left axis represents the anomaly of total water storage, while the right axis represents the absolute total water storage. While the simulated value of total water storage is dependent on its initial value, the fluctuations of this value are independent of its initial setting. Therefore, this comparison is intended to demonstrate that our model effectively captures the dynamic changes in total water storage.

---

## Author Response (AR1)

**RC1**: ['Comment on hess-2024-7'](), Anonymous Referee #1, 02 Apr 2024
**Review for "System dynamics perspective: lack of long-term endogenous feedback accounts for failure of bucket models to replicate slow hydrological behaviors"**

This paper investigates the factors long-term endogenous feedback soil water-vegetation feedback loop through the system dynamics perspective. It is an interesting topic and is widely concerned. However, there are several problems that needs to be revised.

The products of actual evapotranspiration contain much uncertainties. The authors should be very careful with the products and validation of the accuracy in the region is recommended.

Reply: Thank you for the comment. Two additional products of actual evapotranspiration have been included for cross-validation the ETWatch and NTSG products. The text has been added in Method section 2.2 spanning lines 140 to 150 as below:

"*Furthermore, two alternative AET products, CR (Complementary-Relationship-based modelling, Ma et al., 2019a,b) and PEW (Proportionality hypothesis-based surface Energy-Water balance model, Fu et al., 2022a,b) were taken for cross-validation with ETWatch and NTSG. Complementary-relationship and Priestley-Taylor approaches were employed by CR and PEW, respectively. Despite their algorithmic difference, the four AET products exhibit comparable values and common characteristics (Fig. S2). Firstly, all products successfully captured the annual AET fluctuations. Secondly, in terms of long-term trends, all products indicated a decline in AET at the end of 1990s and the beginning of 2000s. While AET data of NTSG and ETWatch showed an upward trend thereafter, AET data of CR and PEW continued to decrease during the study period. Thirdly, the magnitude of fluctuations of P and Q reduced in the later stage. Given these observations, we conclude that the ETWatch product demonstrates reliable values and the most distinguishable pattern among the four AET products, thus deemed reliable choice for AET estimation.*"

There is a hydrological model adopted. How the parameters are determined, including VEG and GP? Why the model is adopted? To my knowledge, process-based hydrological model is rarely used at annual scale.

Reply: Thank you for the comment. Firstly, the explanation about how the VEG and GP parameters determined has been added in Result section 3.3 spanning lines 359 to 367 as below:

"*Both observed data and literature review were used to set up human activities scenarios. Firstly, given the lack of groundwater pumpage data for the study area and*

*the time frame, we estimated it based on existing literature. The water use of household sector in FP county mainly relies on groundwater and water demand is projected to increase in the future (Wei and Wang, 2019). Furthermore, the groundwater depletion or the decline of water table in North China Plain also showed an exponential increase during the study period (Gong et al., 2018; Lancia et al., 2022; Huang et al., 2023). Secondly, NDVI data showed a linear upward trend during the majority of the study period (Fig. S9), and literature showed that the Taihang Mountain Afforestation Project launched in 1987 and the coverage of vegetation increased 11.2% from 1994 to 2013 (Hu et al., 2017, 2019; Yuan et al., 2019).*"

Secondly, different from conventional hydrological models, our model is a system dynamics model which primarily emphasizes the interplay between system components. In system dynamics models, the choice of time step is dictated by how the state variables update over time according to your purpose, rather than the infinitesimal time intervals used in traditional numerical models. Selecting an appropriate time step is crucial for yielding an acceptable approximation of system dynamics for a given purpose.

The hydrological mechanism of annual time step has been inspired by a previous study which result showed that a clayey-soil catchment tend to exhibit higher flow in the short term but less discharge in the long term than its sandstone counterpart (Xiao et al., 2019). Here, "short-term" refers to time scale of hours to days, while "long-term" refers to annual time scale. This observation aligns with other studies indicating that the mean travel time of a non-groundwater-dominant catchment is around one year or slightly longer (Sterte et al., 2021). These findings prove that catchment soil water stock typically updates at an annual step, and highlight the distinct mechanisms underlying the short-term and long-term hydrological behaviors.

So a sentence was added in Method section 2.3.3 spanning lines 263 to 265 as "*The choice of the time interval is due to both the study's purpose and the fact that mean travel time of a non-groundwater-dominant catchment is around one year or slightly longer, leading to catchment soil water stock typically updates at an annual step (Sterte et al., 2021).*"

Xiao, D., Shi, Y., Brantley, S.L., Forsythe, B., DiBiase, R., Davis, K., and Li, L., 2019. Streamflow generation from catchments of contrasting lithologies: the role of soil properties, topography, and catchment size. Water Resources Research, 55, 9234-9257.

Sterte, E.J., Lidman, F., Lindborg, E., Sjoberg, Y., and Laudon, H., 2021. How catchment characteristics influence hydrological pathways and travel time in a boreal landscape. Hydrology and Earth System Sciences, 25, 2133-2158.

The desired (expected) soil moisture stock ESMS is key parameter in the hydrological model. The meaning of ESMS need to be further explained. The discussion in Section 4.1 seems too general.

Reply: Thank you for the comment. More discussion about the ESMS parameter has been added in Discussion section 4.2 spanning lines 454 to 471 as below:

"*Secondly, ESMS as an important parameter represents the soil water holding capacity with value ranging from field capacity to wilting point. Thus, accurate estimation of ESMS is crucial for gaining insights into the long-term hydrological behaviors. However, ESMS estimation remains a significant challenge in hydrology, primarily due to the complexity of measuring the total volume of soil capillary pores across an entire catchment. The measurement is difficult using the "bottom-up" method as underground characteristics, such as soil depth and soil porosity, exhibit high heterogeneity and are not readily observable. Moreover, such measurements are often labour-intensive, constrained by space and depth limitations. In our study area, soil characteristics are sparsely observed (Fu et al., 2021), and the results obtained from limited soil samples have a strong bias against the plausible distribution. Furthermore, soil structural characteristics undergo gradual changes over time due to the intricate interplay of soil physical, chemical and biological components. Often, the measurement scales are significantly smaller than the scales relevant to changes in soil structural characteristics.*
*To address these challenges, this study introduces a "top-down" methodology for estimating ESMS, grounded in system dynamics principles. Following the exogenous interventions, a dynamical system evolves over time, with its trajectories repeatedly passing through a fixed point or multiple points called "attractor" (Rickles et al., 2007). In this context, ESMS can be considered as "attractor" of the hydrological system, which can be inferred from the trajectories of hydrological system dynamics. Previous studies have also suggested that vegetation metrices, such Gross Primary Productivity (GPP), Leaf Area Index (LAI), can serve as surrogates for soil structure modifications and soil hydraulic properties (Jha et al., 2023). Although vegetation is still considered as exogenous driver in that framework, they highlight the intricate interactions among vegetation, soil structure, and soil moisture.*"

Fig 4. What's the meaning of model Z. Could the causal relationship be used to generate better hydrological series compared the the process-based hydrological model?

Reply: Thank you for the comment. Model Z is actually the autoregressive model in Granger's Causality Test and the y-axis is the predicted values from the model. We have revised the code-auto-produced label and changed "model Z" to the name of predicted variables.

Future dynamics of hydrological systems under three climatic scenarios are shown in Fig 7. But could not catch what the key points the author wanted to address.

Reply: Thank you for the comment. The figure has been updated to a new version that contrasts our results with CMIP6 results (as illustrated below). In Method section 2.2, the acquirement and process of CMIP6 data have been described spanning lines 162 to 175 as below:

"*Future monthly rainfall ("pr" in CMIP), monthly total runoff ("mrro" in CMIP) and AET ("evspsblsoil" and "evspsblveg" in CMIP), stemming from three scenarios (SSP126, SSP245, and SSP585) within four Global Climate Models (ACCESS-CM2, CNRM-ESM2, EC-Earth3, and GFDL-CM4), were retrieved from the CMIP6 (the phase 6 of the Coupled Model Intercomparison Project, https://esgf-node.llnl.gov/search/cmip6/), covering the time span from 2015 to 2100. Notably, the GFDL-CM4 model lacks the SSP126 experiment, and the ACCESS-CM2 model lacks the "evspsblveg" variable. The spatial resolution varies, with 250 km for ACCESS-CM2 and CNRM-ESM2 models, and 100 km for EC-Earth2 and GFDL-CM4 models. The grid values corresponding to the study area were extracted. For ACCESS-CM2 (CNRM-ESM2), the grid value at the 62nd (83rd) row and 104th (93rd) column, corresponding to longitudes of 115.3125° (115.3125°) and latitudes of 39.375° (39.9218°), were used. Similarly, for EC-Earth3 (GFDL-CM4), the grid values at the 164th (92nd) and 165th (93rd) rows, and the 185th (130th) column, corresponding to longitudes of 114.6094° (114.375°) and 115.3125° (115.625°), and latitudes of 39.649° (39.5°), were utilized. The unit of kg m-2 s-1 was converted to mm month-1 by multiplying it with 86400 seconds and 30 days. Subsequently, annual values were calculated by summing the monthly values. The annual rainfall was averaged from four GCMs and used to drive system dynamics (SD) model, generating Q and AET estimates under different SSP scenarios.*"

And in Result section 3.3, the comparison results have been described spanning lines 386 to 394 as below:

"*Using the system dynamics' structure, we computed future runoff and AET from 2015 to 2100 under different SSP+RCP scenarios and compared the results with those from four GCMs. In term of AET, the system dynamics' structure aptly captured its primary behavioral patterns: a decline in 2040s and 2070s, followed by an increase in 2060s and 2090s. Notably, the R-squared values between system dynamics' results and GCMs' results were relatively higher for SSP585, specifically 0.48 for ACCESS-CM2, 0.15 for CNRM-ESM2, 0.41 for EC-Earth2, but 0.04 for GFDL-CM4. For the other two SSPs, the R-squared values were relatively poor. Regarding runoff, the system dynamics' simulations indicated an increasing trend, whereas the GCMs' estimation did not exhibit any discernible trend. The R-squared values between system dynamics' results and GCMs' results for runoff were poor across all experiments. These findings further emphasized that process-based models, while adept at capturing short-term fluctuations, may be inadequate in simulating long-term hydrological behaviors.*"

[Figure]

A general description of the hydrological properties of the catchments is needed.

Reply: Thank you for the comment. The vegetation information has been described in the manuscript, soil properties are added in Method section 2.1 spanning lines 110 to 116 to further illustrate the hydrological properties.

"*Rainfall is the main source of streamflow in the study area (Fu et al., 2024). The main geologic characteristics of the Taihang Mountain area are "soil and rock dual texture", with thin topsoil of 0.2-0.5 m rich in plant roots and gravel, and a thick 0.5-10 m subsoil of weathered granite gneiss with highly developed fractures (Cao et al., 2022). The soil of Taihang Mountains is primarily developed from granite, gneiss, limestone, and sandstone. Cambisols and luvisols constitute the dominant soil types, accounting for 46.86% and 15.46% of the total area, respectively (Fu et al., 2021). These soil types all have high content of sand gradation, approximately 50%, followed by silt. Clay comprises the smallest proportion of the soil content, at approximately 20% (Yang and Cao, 2021).*"

The use of "bucket models" in the title seems inappropriate.

Reply: Thank you for the comment. We have changed the title into "*How system dynamics explains long-term hydrological behaviors? The role of endogenous linkage structure*". Please see the first reply to reviewer 2 for the change.

Line 116. Using ΔS to indicate water budget seems not suitable.

Reply: Thank you for the comment. We have replaced all the "*water budget*" with "*catchment water storage change*".

Lines 208-211. It is inappropriate to put the discussion there.

Reply: Thank you for the comment. These sentences have been expanded to explain the division of climatic phases in Result section 3.1 spanning lines 271 to 280 as below:

"*Here we divide the whole study period into two distinct phases — wet and dry. Firstly, from climatic perspective, this is because, the PDO phase shifted from positive to negative mode since the late 1990s, corresponding to the decrease in rainfall in the study area (Fig. S6). Meanwhile the self-calibrating Palmer Drought Severity Index (SC-PDSI) and the Standardized Precipitation Evapotranspiration Index (SPEI) calculated from the climatic data in Beijing, China, both showed major dry periods in 1980-1985 and 1999-2011 (Zhao et al., 2017). Secondly, in term of streamflow, streamflow of the headwater of Baiyang Lake significantly decreased, with tipping point in around 2000 (Fig. S7; Cui et al., 2019; Xu et al., 2019), although precipitation does not show significant trend (Fig. S8). Thirdly according to wavelet analysis, a dampening process in the magnitude of fluctuations in the wavelet coefficients was identified for P and Q after 2000. All these information indicated that the hydrological system has transitioned from wet phase to dry phase around 2000.*"

Lines 210-214. The brief description of hydrological characteristics of the wet and dry phase is suggested. Why the wet-dry phase be determined by wavelet coefficient instead of annual precipitation or aridity index?

Reply: Thank you for the comment. Please see the former response for the explanation of the division of climatic phases.

Fig 2. Subtitle of the figure needs to be labelled and also for other figures.

Reply: Thank you for the comment. We have labelled the figure subtitles for Figure 2 and other figures.

Fig 5. What's the meaning of the three sub-figures for Fig 5(a).

Reply: Thank you for the comment. The three sub-figures of Fig 5(a) is a step-by-step process to derive causal structure of hydrological system. The figure has been revised and more explanations have been added in main text and figure caption.

Lines 327-328. Penman-Monteith model is used for calculating the potential ET instead of actual ET.

Reply: Thank you for the comment. The formulation of "Penman-Monteith" has been deleted in our revised manuscript. Here we would like to explain it a little bit more. Reviewing the derivation of the Penman-Monteith (PM) equation is instrumental in understanding how it functions. Initially, Penman integrated the surface energy balance equation with the aerodynamic equation to formulate the Penman equation for estimating evaporation (Dolman et al., 2014). Later, Monteith improved on the Penman equation by incorporating a surface resistance term and a more rigorous term for aerodynamic transfer (Allen, 2005). Therefore, the PM equation originally allowed the direct estimation of evaporation from unsaturated surface by utilizing actual thermodynamics parameters. Subsequently, the FAO56 modified PM equation, transitioning from a one-step approach to the two-step $K_c$-$ET_0$ approach, thereby enhancing its global applicability.

Dolman, A.J., Miralles, D.G., and de Jeu, R.A.M., 2014. Fifty years since Monteith's 1965 seminal paper: the emergence of global ecohydrology. Ecohydrology, 7(3), 897-902.

Allen, 2005. Penman-Monteith equation. Encyclopedia of Soils in the Environment, Elsevier, Academic Press, Pages 180-188.

Line 331. What's the specific of instantaneous scale?

Reply: Thank you for the comment. The formulation of "instantaneous scale" has been deleted in our revised manuscript. Please see the Discussion section 4.1 for more details about the scale issue.

**RC2**: ['Comment on hess-2024-7'](), Anonymous Referee #2, 18 May 2024

The study uses a system dynamics approach aiming to reveal endogenous factors that account for the long-term slow dynamics of a catchment hydrological system. The paper addresses relevant problem within the scope of HESS, the topic is interesting and widely discussed. However, my recommendation is major revision due to several significant issues in the methodology that seem inadequate for addressing the posed research questions. Further justification is also necessary for the conclusions made. Below are my main concerns.

First, I did not find confirmation that the studied basin demonstrates "slow hydrological behaviors." If I understand correctly, the authors believe that such confirmation is provided by the obtained results of wavelet analysis. I do not think so. Wavelet analysis evaluates the coincidence or difference in the phases of oscillations of various hydrological variables, which can be useful for assessing the cause-and-effect relationships between them as well as for identifying long-term wet-dry periods, as was shown in the paper. But the results presented, in particular the fact of a lag in the response of evapotranspiration to precipitation, do not indicate per se the slow behavior of the system. Such evidence could follow from an analysis of the presence of a long-memory effect in the studied hydrological time series. (Note, that in a paper devoted to the problems of detecting slow hydrological behaviors and the physical mechanisms that control this behavior, it would be appropriate to reference at least the most well-known hydrological publications in this area (e.g., some of them cited by O'Connell et al. (2016)). However, the results of the description of hydrological processes in the studied catchments using standard autoregressive models presented in the paper (Fig. 4) allow us to doubt that these are long-memory processes.

Reply: Thank you for the comment and encouragement. To address this problem, first we gave a clear definition about hierarchies and scales in Introduction section spanning lines 94 to 104 in our revised manuscript as below:

"*To make the discussion clearer, the "long-term" is constrained. As a complex system, a hydrological system comprises multiple hierarchies. Each hierarchy is governed by a distinct mechanism and produces fluctuations/cycles at certain time scale. Notably, the time series yielded at each hierarch can be considered as "long-term" or "slow" compared to its predecessor. Here, we define that the low hierarchy corresponds to sub-year scale, the high hierarchy corresponds to interannual scale, and the even higher hierarchy corresponds to decadal scale. This definition has been inspired by a previous study which result showed that a clayey-soil catchment tend to exhibit higher flow in the short term but less discharge in the long term than its sandstone counterpart, indicating distinct underlying mechanisms to control streamflow generation at different hierarchies/time scales (Xiao et al., 2019), as well as a study which proposed a fill-spill phenomena across scales (McDonnell et al., 2021). Considering the objectives of this study and the data span, we specify the "long-term" as interannual to decadal scales.*"

Then we discussed the hydrological mechanisms of different hierarchies and scales in Discussion section 4.1 spanning lines 401 to 424 as below:

"*Here we discuss the interpretation of the underlying mechanisms in term of the hierarchies we defined in hydrological system. At the lowest hierarchy, hydrological processes are primarily controlled by the interaction between rainfall density and sink-filling/macropore flow (McDonnell et al., 2021), which was conceptualized as a small, active reservoir (Xiao et al., 2019). At this level, soil properties, particularly non-capillary pores in the top layer, are predominant and direct factor influencing the storage-discharge relationship at intra-annual scale. At the higher hierarchy, hydrological processes primarily involve the interaction between vegetation structure change and soil bound water, defined as water stored in capillary pores (Good et al., 2015), which was conceptualized as a large, passive reservoir (Xiao et al., 2019). At this level, soil capillary pores in the deep layer become predominant and direct influencing factor. Water stored in capillary pores is primarily utilized by vegetation, further strengthening the link between vegetation and hydrological behaviors at the interannual scale. As ascending to even higher hierarchy, the interaction between climatic oscillation and soil water holding capacity takes over the priority of a hydrological system to generate decadal-scale hydrological dynamics. This is because, soil water holding capacity can be enhanced by the interplays of soil physical, chemical and biological components during drought (Delgado and Gómez, 2016). Physically, drought alters aggregate size fraction with less macro- and more micro-aggregates (Zhang et al., 2019; Su et al., 2020). Also pore network connectivity decreases due to reduced vegetation growth (Nagassa et al., 2015), which likely increases soil water holding capacity under drought conditions because of poorly connected soil pore network and larger water holding capacity of micropores (Patel et al., 2021). Chemically, the ionic strength of soil solution can be up to nine orders of magnitude greater in dry soils compared to saturated soils, creating extremely different chemical environment (Patel et al., 2021). This can cause desorption of different carbon molecules from minerals, and soil organic carbon (SOM) destabilization (Bailey et al., 2019). Biologically, drought stress reduces population and activity of soil animals and microbes. This can also destabilize SOM because the microbial community would preferentially consume the more labile/biodegradable organic molecules, leaving behind a more uniform, stable SOM pool (Kaiser et al., 2015). Furthermore, rainfall pulses in drought period would drive water going deeper because rainfall in dry spells usually occurs as discrete events (Manzoni et al., 2020). These make water easier to be trapped by soil particles rather than utilized by plants, further decreasing evapotranspiration.*"

Second, if there is evidence that the dynamics of the system can be interpreted in the desired way, then the use of standard wavelet analysis, which was developed for processes with short memory, is questionable. For such processes, the analysis has to be modified (see, e.g., Percival and Guttorp, 1994; Hsu, 2006).

Reply: Thank you for the comment. As previous response, this paper aims to elucidate the driving mechanism of hydrological cycles at interannual to decadal scales. Wavelet analysis is a commonly used tool to identify these varying cycles. We will change the words of "long"/"short" to specific time scales.

Third, as an alternative that allows one, in contrast to the bucket model, to describe "slow hydrological behaviors," a simple linear model of annual changes in the components of the water balance of the river basin under study is proposed. In the paper, I was unable to find results demonstrating that such a model has an advantage in describing the slow dynamics of a system, so I see no reason to consider the proposed model a reasonable alternative. The calculation results for ET, Q and TWS (Fig. 6 and Figs. in Suppl. Materials) are poor, i.e. the suggested model not only does not reproduce the desired effects, but also does not meet the performance measures adopted for hydrological models (the values of the coefficient of determination given in the paper confirm my opinion).

Reply: Thank you for the comment. The discussion of the advantages of our model has been added in Discussion section 4.2 spanning lines 440 to 453 as below:

"*Firstly, the system dynamics' model in our study is a simplified representation of the complex reality. We argue that both scientific and developmental merits are involved in the proposed model. In essence, the model is a "toy" model, also known as a minimal or exploratory model. Toy models are used in theoretical physics and many advanced research scopes (Georgescu, 2012). Complex systems usually involve a vast number of interacting elements; however, it is often the case that a small number of factors are quantitatively more significant than the others. Therefore, our highly simplified model only includes key factors providing a good starting point for building a more complex model (Luczak, 2017). Toy model can help us understand a complex system in its broadest strokes. By breaking it down and then building it back up, we gain profound insights into the system's essence. Moreover, although the lack of short-term hydrological behaviors may reduce the performance of our model, it is not significantly inferior compared to large models. For instance, previous studies evaluated runoff simulation from global climate (GCMs), global hydrological (GHMs) and land surface models (LSMs) and results showed that the GCMs failed in capturing observed runoff with median value of Pearsons correlation coefficient (r) close to 0, while GHMs and LSMs can capture observed runoff with a median value of r higher than 0.6 (r-square values ranging from 0.5-0.6) (Zhou et al., 2012; Hou et al., 2023). Given this, our model (r-square values around 0.3) exhibits comparable performance to large models, while requiring significantly less data and computational time.*"

Thus, I have to conclude that the title of the paper does not reflect the content of its current version, since it does not provide any evidence that "lack of long-term endogenous feedback accounts for failure of bucket models to replicate slow hydrological behaviors."

Reply: Thank you for the comment. We have changed our title into "*How system dynamics explains long-term hydrological behaviors? The role of endogenous linkage structure*".

Nan-Jung Hsu (2006) Long-memory wavelet models. Statistica Sinica 16, 1255-1271

O'Connell P.E. et al. (2016) The scientific legacy of Harold Edwin Hurst (1880–1978), Hydrological Sciences Journal, 61:9, 1571-1590, DOI: 10.1080/02626667.2015.1125998

Percival, D.B. and Guttorp, P. (1994) Long-Memory Processes, the Allan Variance and Wavelets, Editor(s): Efi Foufoula-Georgiou, Praveen Kumar, Wavelet Analysis and Its Applications, Academic Press, 4, 325-344

**Specific and technical comments**

1. Lines 148-151: This describes the Granger's Causality Test for determining the dependence of X on Y (Y causes or does not cause X), while equation (2) describes the dependence of Y on X.Change either the description on lines 148-151 or equation (2).

   Reply: Thank you for the comment. We have changed the description accordingly.

2. Line 239: If ET Granger-causes S, and ΔS Granger-causes ET, then the question is: what is the cause and what is the effect?

   Reply: Thank you for the comment. More explanation about the bidirectional causal relationship has been added in Result section 3.2 spanning lines 307 to 311 as below:

   "*The bidirectional causal relationship between ΔS and AET denotes the existence of unobserved common variables, referred to as confounders (Pearl, 2009; Morgan and Winship, 2015). According to the theory of causal inference, confounders can obscure or blur the "real" causal relationship, thus it is advisable to establish subgroups to analyze the causal relationship separately. In the context of our study, the causal relationship of ΔS and AET is "confounded" by soil water: AET causes ΔS when soil water is plentiful, whereas ΔS causes AET when soil water is scarce.*"

3. Fig. 4: Change "cause" to "causes".

   Reply: Thank you for the comment. We have changed it accordingly.

4. Fig. 4: What does AR(1) mean? First order autoregressive model? What, then, is AR(1,1)? Explanation required.

Reply: Thank you for the comment. More explanation has been added in caption of Figure 4 as below:

"*AR(p) represents the autoregressive model, where "p" is called the order of the model and represents the number of lagged values we want to include. AR(p,q) represents that additional q-order variable is included.*"

5. Fig. 5: It is not clear what results allowed the authors to conclude that reinforcing feedback (1) exists.

Reply: Thank you for the comment. We have explained that the prior knowledge is needed for derivation of reinforcing feedback in Method section 2.3 spanning lines 177 to 184 as below:

"*We propose three approaches — wavelet analysis, Granger's causality test and system dynamics. Although some information about the endogenous linkage structure of hydrological system can be inferred from the wavelet analysis and the Granger's causality test, additional knowledge from multiple subjects is necessary to build the system dynamics' structure. For example, considering vegetation growth as a reinforcing feedback loop requires collective prior knowledge of plant physiology (Lian et al., 2021; Wright and Francia, 2024), resource competition theory (Craine and Dybzinski, 2013), population ecology (Snider and Brimlow, 2013), and likely other knowledge not included in this study. Therefore, the proposed three approaches run in parallel but are not sequential and help us understanding the endogenous linkage structure among the stocks in the hydrological system.*"

6. Line 257: Until now, there has been no talk about hysteresis effect. Explanation required.

Reply: Thank you for the comment. This sentence has been deleted in our revised manuscript.

7. Lines 268-270: It is necessary to expand the description of the solution to equations (3)-(12). In particular, explain how the variables VEG(t), K(t), ESMS are determined (in the equations, the latter is designated as a constant). How are the constants C1, C2, C3, C4 determined?

Reply: Thank you for the comment. The initial values of these parameters are first obtained from literature, then calibrated against observed ET and discharge.

8. Lines 271-272: The presented results do not confirm the statement that "Simulated Q and $\Delta$S captured both the annual fluctuations and the long-term trends".

Reply: Thank you for the comment. We modified the sentence with specific time scales added.

9. Fig. 6 and Figs. in Suppl. Materials: The calculation results for ET, Q and TWS are poor. Herewith, good results were obtained for ΔS. Why? I believe that these results are achieved by adjusting the calibration factors and variables. In this case, coefficients C1-C4 have no physical content and can take on any values. For ΔS, this adjustment made it possible to compensate for the poor calculation results of other variables. Please comment.

Reply: Thank you for the comment. As mentioned above, we calibrated our model against observed ET and Q but not ΔS. The high accuracy of ΔS is achieved due to the slow rate of change in soil water, which is compatible with the time scale we are primarily focusing on. Moreover, the physical meaning of coefficients C1-C4 have been added in Method section 2.3.3 spanning lines 260 to 262 as below:

"*C1, C2, C3, and C4 are dimensionless coefficients with physical interpretations of respectively, proportion of impermeable area, proportion of soil bound water utilization by AET, proportion of rainfall into deeper layer, and proportion of outflowing groundwater.*"

10. Fig. 6: What is ET1, ET2, Q1, Q2, ΔS1, ΔS2, TWS1, TWS2? What is TWSA?

Reply: Thank you for the comments. The further explanation has been added in figure caption as below:

"*AET1, Q1, ΔS1 and TWS1 represent the simulated values with constant ESMS, while AET2, Q2, ΔS2 and TWS2 represent the simulated values with varying ESMS. TWSA is the anomaly of total water storage.*"

11. Lines 298-302: How were anthropogenic impacts (VEG, GP) set for the future period? How were coefficients C1-C4 set? The same as for the historical period? On what basis, if these are purely empirical coefficients reflecting data for the observation period? Overall, I see no point in using an ineffective hydrological model to estimate the future state of a hydrological system. In addition, these experiments are not relevant to the main content of the paper. I suggest removing them.

Reply: Thank you for the comment. Please refers to the reply to the 9th specific comments. Furthermore, we temporarily set VEG and GP as 0 and fix C1-C4 in the future projection. These parameters may change over time. In this study, however, we only show the simplest model and want to capture the essence of hydrological dynamics at multi-year to decadal scales. In the future

work, with more details added in, these parameters can be changeable to get more accurate results.

12. The reasoning in subsection 4.1 is correct in essence but has no relation to the results obtained. The listed physical, chemical, and biological mechanisms influencing changes in soil water retention are not described by the extremely simple model proposed. Therefore, the fact that the desired soil moisture turned out to be higher in the dry phase has nothing to do with these mechanisms but, as I assume, is only an accidental consequence of the calibration procedure.

    Reply: Thank you for the comment. Please refers to the reply to the first comment. The slow shift of soil water holding capacity with the change of capillary pores, and its influence on hydrological cycles have been captured by our model due to the improvement of model performance with higher ESMS in the dry phase.

13. Also, the reasoning in subsection 4.2 is not relevant to the results obtained. The previous sections do not show that vegetation changes, "such as tree growth and mortality, have become significant factors influencing ET over climate". It is not clear what long-term delay in hydrological response the authors are talking about. No results were presented to support the presence of such relationships. The conclusion that "persistent hydrological shifts and especially flow reductions such as those caused by the increasing and enduring multi-year drought can only be described accurately with the system dynamics approach" is not supported by the presented results.

    Reply: Thank you for the comment. According to the reply to the first comment, we will reorganize the discussion and focus on the different hierarchies and scales that our model at. We believe our study contributes to the research gap identified by Fowler et al. (2020) on improved understanding of the hydrological dynamics at interannual to decadal scales.

**RC3**: 'Comment on hess-2024-7', Anonymous Referee #3, 22 May 2024

This study seeks to look at the problem of "slow hydrological behaviours" via a different lens, namely that of system dynamics. I certainly agree that this is a pressing problem that needs some fresh thinking, since many current "bucket" models do a poor job at simulating such behaviours. The system dynamics approach may be relevant here, but I'm concerned that there are multiple serious issues with the implementation here, as follows.

- **Missing processes, and conceptual confusion, regarding evapotranspiration.** The authors don't seem to be taking into account the concept of evaporative demand - that is, they are neglecting the fact that the actual ET is a function of two things: (1) how hot and dry the near-surface atmosphere is, as captured in the concept of Potential Evapotranspiration (PET); and (2) the subsurface water availability, eg. in soil moisture. (1) seems to be missing so let's focus on that. At line 177 it says "while traditional hydrological models usually use physical process-based models such as the Penman or the Thornthwaite models to calculate ET...". [NB: By "ET", I believe the authors here mean actual evapotranspiration (AET), because this is how they have used that acronym throughout the manuscript - which is part of the confusion.] Anyway, no, that is not what "traditional" models do - typically, methods such as Penman are used to estimate PET, not AET; such methods produce a PET timeseries which becomes an \*input\* to the modelling processes. In other words, whereas the modelling here seems to use only one input, precipitation, traditional modelling uses two, precipitation and PET. So the present method seems to be neglecting the reality that a hotter drier atmosphere can result in a greater proportion of precipitation being lost to AET, all else being equal. I presume the same framework could be altered to add this additional driving variable, but I'm unsure. In any case, the modelling is subsequently applied to a climate change scenario which invariably means a hotter (if not drier) world, and yet the modelling seemingly cannot account for one of the most basic elements of the climate change signal (ie. rising temperatures)—this is unacceptable. I note that I may have misunderstood something here so I'd be happy to be corrected by the authors.

Reply: Thank you for the comment. First, the discussion about evaporative demand has been added in Discussion section 4.2 spanning lines 472 to 480 as below:

"*Finally, it is notable that our model was developed for a study area characterized by a semi-arid climate with limited water resources. In such regions, energy (temperature and light) is typically abundant and does not constrain vegetation growth or AET. Despite the higher evaporative demand resulting from hotter and drier atmospheric conditions, lack of water hinders the increase in AET by drying out the soil surface, weakening the vegetation, or even causing vegetation death to decrease AET. Therefore, the energy factors such as temperature and light are omitted in our model as our model aims to capture the most critical mechanism in the study area. The success of the model in replicating nonlinear hydrological dynamics*

*proves that the core factors has been identified. For regions with energy limitation, such as high latitudes and tropic regions, energy factors could serve as the primary factors and the soil water-vegetation interaction should be modified into the interaction between energy (temperature/light) and vegetation.*"

Second, the subsurface water availability is one of core factors in our model, which implicitly included in our time scales. Please see the replies to reviewer 1 and 2 for detailed description.

We apologize for any lack of clarity in explaining the background and context of our work. The above discussion will be integrated into Discussion section.

- **The chosen approach induces memory that is too long for some system components**.   The calculation of key fluxes (AET, Q, recharge) depends upon the discrepancy, DISC, between soil moisture and the assumed ideal or "expected" soil moisture storage. The trouble is that since the equations are being solved at a yearly timestep, this means the AET, Q and recharge don't respond to a high rainfall value until the year after it happens.   Basic common sense would seem to ward against this - ie. the flood doesn't come the year after the rainstorm.   I wonder if this could be solved if the same approach were used on a shorter time step and/or if the equations were solved in such a way that the equations took account of forcing fluxes from the same timestep than the one being solved.

Reply: Thank you for the comment. Please refer to the first reply to reviewer 2. Hydrological system comprises multiple hierarchies, with each hierarchy characterized by a distinct primary driving mechanism that generates time series fluctuating at different time scales. Our study focus on hydrological dynamics occurring at interannual to decadal scales. Specifically, the interaction between soil bound water and vegetation growth is dominant at the multi-year scale, whereas the interaction between climate oscillation and soil water holding capacity is dominant at the decadal scale. In addition, our model, albeit the minimal structure in design, serves as a conceptual framework to elucidate the fundamental hydrological dynamics at multi-year to decadal scales. We believe that in the future, as our model evolves to incorporate more intricate details, additional mechanisms and factors will emerge to enhance our understanding of hydrological dynamics across the various hierarchies.

- **The conceptualisation is seemingly ill-suited in the case of streamflow.**  The observed peaks in streamflow are absent, associated with severly underestimated streamflow variability.   The streamflow is calculated as the aforementioned discrepancy in soil moisture multiplied by a calibratable parameter.   Since soil moisture is a state variable that is only allowed to vary on annual timesteps, it is rather slow moving, not episodic, and this property is then translated to streamflow dynamics as well.

Reply: Thank you for the comment. Please see the reply to previous comment and reviewer 2. You precisely capture the key point in our model and we have discussed it in the discussion section.

MINOR COMMENTS:

Line 109: I suggest that "AET" is used instead of "ET" throughout - this will minimise confusion with PET

Reply: Thank you for the comment. All the "ET" has changed to "AET" according to the comment.

117: It would be good to discuss the uncertainty of information being used here, particularly remotely sensed AET. Given the uncertainty in the RS AET, it's ill advised to calculate change in S by subtracting RS AET from precip (for an example of how a study accounted for this uncertainty by water-balance-based factoring, see here: https://doi.org/10.1029/2022WR033538).

Reply: Thank you for the comment. Please refers to the reply to reviewer 1. We have downloaded three additional AET products and compared them with the ETWatch product. The four AET products are produced using different algorithm. Generally, they are equivalent in values but do show uncertainties in fluctuations and trends. Among them, the ETWatch product stands out for displaying the most prominent fluctuations. This comparison has been added in Method section.

136: Many readers will not have a background in the concepts and/or methods used. Please explain the concept of a mother wavelet.

Reply: Thank you for the comment. More explanation about mother wavelet has been added in Method section 2.3.1 spanning lines 194 to 196 as below:

"$\Psi$ is known as "mother" wavelet because it can generate "child" wavelets by dilation and translation. Function is then processed with these child wavelets to yield wavelet coefficient (Sayood, 2012)."

In general, much of the language used anthropomorphises the system, such as saying it has a "goal" ("[balancing feedback loops are] aiming for stability"). Also, where it is said that it "desires" a certain soil moisture.

Reply: Thank you for the comment. The "desired" is used in system dynamics due to it is often used to simulate economic dynamics. For a balancing feedback loop, its "goal" is to keep system stable by approaching the "desired" value.

196: Better to call it "precipitation input" rather than "precipitation inflow" as the latter is too close to language used for streamflow. Likewise the other "outflows" would be better as "outputs".

Reply: Thank you for the comment. The "inflow" and "outflow" are typically used in system dynamics and correspond to "stock". We believe that the nonlinear dynamics can be better implicitly represented by these terms.

200: "Correction coefficients" makes it sound like something is incorrect. Perhaps "coefficients that determine the rapidity with which the system self-corrects after a disturbance" or "responds after a disturbance"

Reply: Thank you for the comment. We have deleted the "correction" and the physical meaning of the coefficients have been added in Method section 2.3.3 spanning lines 260 to 262 as below:

"*C1, C2, C3, and C4 are dimensionless coefficients with physical interpretations of respectively, proportion of impermeable area, proportion of soil bound water utilization by AET, proportion of rainfall into deeper layer, and proportion of outflowing groundwater.*"

215: This is method and needs to be moved to the methods section.

Reply: Thank you for the comment. This sentence has been moved to method section 2.3.1 accordingly.

Figure 2. The choice of colour is confusing - Greens and blues are typically reserved for fluxes like precip and streamflow, while reds and yellows for ET. So, swap colour of P and ET.

Reply: Thank you for the comment. The figure has been revised as this suggestion.

252 onwards: There needs to be a subsection in the methods section that explains the broad logic of doing this (even though it won't yet be possible to be specific since the results are yet to be presented). That is, you need to say that you build the model based on the causal links seen in separate analysis.

Reply: Thank you for the comment. A paragraph to explain the relationship among the three approaches has been added in Method section 2.3 spanning lines 177 to 184 as below:

"*We propose three approaches — wavelet analysis, Granger's causality test and system dynamics. Although some information about the endogenous linkage structure of hydrological system can be inferred from the wavelet analysis and the Granger's causality test, additional knowledge from multiple subjects is necessary to build the*

*system dynamics' structure. For example, considering vegetation growth as a reinforcing feedback loop requires collective prior knowledge of plant physiology (Lian et al., 2021; Wright and Francia, 2024), resource competition theory (Craine and Dybzinski, 2013), population ecology (Snider and Brimlow, 2013), and likely other knowledge not included in this study. Therefore, the proposed three approaches run in parallel but are not sequential and help us understanding the endogenous linkage structure among the stocks in the hydrological system.*"

264: the remotely-sensed vegetation indices are discussed as if the authors are unaware that remotely-sensed AET is itself calculated from one of these indices (eg. NDVI)

Reply: Thank you for the comment. The sentence has been revised spanning lines 340 to 343 as below:

"*Currently, the AET estimation is based on remote-sensing-based vegetation index, however, which makes the predictions of future AET impossible because the future vegetation index cannot be obtained.*"

301-02: No, that's not the way to interpret GCM sequences because, although it is hoped that their climate sequences are realistic overall (ie. have similar statistics to reality for the sequence pre-2024), the exact timing of dry periods and wet periods is decoupled from reality; ie. the simulated sequence is purely synthetic (thus, the droughts that appear in the pre-2024 simulated sequence are not timed with historic droughts).

Reply: Thank you for the comment. Please see the reply to reviewer 1. Figure 7 has changed into a comparative analysis of the results between our model and 4 Global Climate Models (GCMs). This comparison reveals both consistencies and inconsistencies. Interestingly, the projected runoff by GCMs does not show long-term trend, while our model predicts a gradual increase in runoff in the future.

Figure 6: the bottom right hand panel is cheating a bit: while it's ok to shift the lines vertically to emphasise the match, the rate of change per unit length should be identical. In other words, since the axis range on the right is 2000 (ie. simulated TWSA has maximum 2000, minimum 0), the other axis (for observed TWSA) should have a range of 2000 also (ie. perhaps -500 to +1500). The current arrangement artificially inflates the apparent match.

Reply: Thank you for the comment. The left axis represents the anomaly of total water storage, while the right axis represents the absolute total water storage. While the simulated value of total water storage is dependent on its initial value, the fluctuations of this value are independent of its initial setting. Therefore, this comparison is intended to demonstrate that our model effectively captures the dynamic changes in total water storage.

---

## Author Response (AR2)

**Response Letter**

**Report #1**

I thank the authors for their thoughtful and thorough response to my criticism. The revised version of the manuscript satisfactorily addresses my principal comments that resulted in improvements to the manuscript in comparison with the previous version.
Reply: Thank you very much for your valuable suggestions and encouragement.

I suggest softening some statements:

L. 89-90 "…the concept has not been used in hydrology under natural conditions". => "…the concept has rarely been used in hydrology under natural conditions"
Reply: Thank you for the suggestion. The sentence has been revised accordingly in Line 91.

L 121 "…no studies investigated the dynamics of the catchment water storage" => … there are not many studies investigated the dynamics of the catchment water storage"
Reply: Thank you for the suggestion. The sentence has been revised accordingly in Line 122-123.

Some papers as an example
Birkel, C.and Tetzlaff, D. (2011). Modelling catchment-scale water storage dynamics: Reconciling dynamic storage with tracer-inferred passive storage. Hydrological Processes. 25. 3924 - 3936. 10.1002/hyp.8201.
Chaffaut, Q. et al (2022) New insights on water storage dynamics in a mountainous catchment from superconducting gravimetry, Geophysical Journal International, Volume 228, Issue 1, January 2022, Pages 432–446, https://doi.org/10.1093/gji/ggab328
Huang, CC. and Yeh, HF. (2022) Evaluation of seasonal catchment dynamic storage components using an analytical streamflow duration curve model. Sustain Environ Res 32, 49. https://doi.org/10.1186/s42834-022-00161-8
Oswald, C et al. (2011). Water storage dynamics and runoff response of a boreal Shield headwater catchment. Hydrological Processes. 25. 3042 - 3060. 10.1002/hyp.8036.

L 501-504
"In conclusion, the system dynamics approach provides a hierarchical view to understand endogenous linkage structure of a hydrological system, and better reproduces the slow hydrological processes at interannual to decadal scales compared to conventional hydrological models" =>
"In conclusion, the system dynamics approach provides a hierarchical view to understand endogenous linkage structure of a hydrological system, and has the potential to better reproduce the slow hydrological processes at interannual to decadal

scales compared to conventional hydrological models

I recommend the manuscript for publication after these technical revisions.

Reply: Thank you for the suggestion. The sentence has been revised accordingly in Line 532-535. Your recommendation is greatly appreciated.

**Report #2**

The authors have gone to some effort to revise their manuscript. However, they have opted merely to discuss the issues raised rather than address them. Given I earlier wrote that the issues raised were serious, I find it disappointing that a more substantive response has not been attempted. I do not feel favourable towards publication unless significant extra analysis is undertaken, as discussed below.

Reply: First, we would like to express our appreciation for your valuable and constructive comments and suggestion. We apologize for any oversight over concerns and comments you have shared with us. We will try our best to address your concerns in this revision.

CORE ISSUE

The main issue I have with the paper is that the authors imply via both words and tone (particularly in the abstract and short summary) that they have improved upon existing methods, yet they have not undertaken a comparison with existing methods - in fact, they have not even demonstrated how to quantify the performance they are interested in. When I pointed out the poor performance on an annual timestep, the defence was that the method is focussed on the multi-annual and multi-decadal timescales. Fine - but the onus is on the authors to find a way to directly quantify this, which has not been attempted. One option could be to separate out the shorter term fluctuations by focus on the residuals around the annual rainfall runoff relationship, which would remove much of the short term fluctuation. Another option is to separate off the low frequency component using, say, empirical mode decomposition or similar. This could be applied in similar fashion to both simulated and observed data, thus isolating the interesting component of the signal and allowing focus on whether it is correctly simulated. The specific technique doesn't matter, the point is that such things are possible, and no attempt has been made. Whatever technique is chosen, it then needs to be applied consistently to allow comparison between their method and a traditional rainfall-runoff model or models.

So, in summary, I would consider the following two things to be a minimum requirement for publication:

1. Quantification of performance on multi-annual to multi-decadal timescales since this is the timescale the authors claim to be focussing on; and

2. Application of (1) to allow a comparison of performance between a traditional rainfall-runoff model and the proposed method.

Reply: Thank you for the comments. We did comparison between our model and a rainfall-runoff model using the empirical mode decomposition method at both

short-term and long-term scales as suggested. As expected, the system dynamics approach outperforms current approaches at long-term scales. Please see the text in line 450-458 in discussion 4.2 and Figure S13 for the revision.

"By adopting the combined structure of a vegetation reinforcing feedback and a soil water-vegetation balancing feedback, our model significantly outperforms traditional rainfall-runoff models and large models at the long-term scale, while demonstrating marginally inferior performance compared to them at the short-term scale. On one hand, a comparison between our model and SIMHYD, a widely used rainfall-runoff model in Australia (Chiew et al., 2002), was conducted in Fuping catchment as an example. Monthly rainfall and potential evapotranspiration were used as inputs to SIMHYD model, and monthly results were aggregated to annual values. Then empirical model decomposition method was applied using Matlab to quantify the performance of both models on short-term and long-term scales (Fig. S13). Results showed that SIMHYD model only surpassed our model in short-term Q simulation, but performed poorly in short-term ΔS simulation and long-term AET, Q, and ΔS simulation."

[Figure]

Figure S13 Comparison between the proposed system dynamics model and traditional rainfall-runoff model SIMHYD on simulating short- and long-term hydrological dynamics in Fuping catchment in China as example. The comparison is conducted by the empirical model decomposition method. OBS is observations, SD is system dynamics model. Here the results of SD model were obtained with varied ESMS (expected soil moisture stock, see eq. 6 in the main text), without VEG and GP which mimic human activity parameters for, respectively, vegetation-related activities such as reforestation, and groundwater pumpage. IMF is the intrinsic mode functions which can be used to extract different resolutions from the data without the use of

fixed functions or filters.

OTHER COMMENTS

In the case of a decision to proceed with publication, the following comments might help to improve the manuscript:

1. AET as input. In response to my suggestion that AET needed to be included as an input, the authors argued that because of the arid nature of the case study, it is not needed. I sense that arguing about this further is not helpful, so I merely request that the manuscript:
- acknowledge that the vast majority of cases will need AET as an input. This needs to be done in the methods and the limitations section of the manuscript, at least.
- confirm that this is possible to include as an input - ie. that the method would allow this.
- write a paragraph aimed at potential users of the method, commenting on how easy or difficult it might be to incorporate AET as an input. Any instructions on how to do this would be welcome, but I recognise that it might be difficult to do this breifly.
Reply: Thank you for the comment. We have integrated vegetation parameter as input in our model, implicitly equating it to the input of AET due to the inherently close correlation between these two factors. A discussion has been added in line 475-482 in discussion 4.2 for the revision.

"On the other hand, the parameter values should be allowed to vary over time rather than being fixed to improve model's performance at the long-term scale. For instance, the fixed utilization ratio of soil water (C2 in Eq. 8) implies that AET solely depends on vegetation coverage and soil bound water, but not on plant species. However, the dominant plant species in the study area have shifted from herbs to shrubs due to the implementation of afforestation projects (Liu et al., 2011), resulting in significant changes in the utilization ratio of soil water (Liu et al., 2014). By incorporating variable parameters, the model's performance can be enhanced, as the input of high-quality vegetation data, including vegetation coverage and plant species over time, is actually equivalent to the input of AET based on the intrinsically close links between the two factors."

2. Given the authors' defence of the poor performance is that their method is focussed on longer timescales, the authors need to discuss options for how to augment their method such that it can also match shorter timescales, otherwise the concept of using it for projection (ie. Figure 7) adds little value. I feel this would fit well with their existing discussion in Section 4.2.
Reply: Thank you for the comment. A discussion has been added in line 465-475 in discussion 4.2 for the revision.

"In the future, our model can be improved from at least two aspects. On one hand, the

mechanism of low hierarchy in the hydrological system can be integrated into the endogenous linking structure to enhance the model's performance at the short-term scale. As discussed earlier, the interaction between rainfall density and sink-filling/macropores controls the hydrological behaviors at intra-annual scale. Thus, high-temporal-resolution rainfall data and detailed soil properties of the topsoil layer are needed to determine the initiation and duration of short-term runoff events. In addition, to link the two hierarchies, intra- and inter-annual together, soil macropores/sinks should be modelled as dynamic features that evolve in response to vegetation changes. Previous studies showed that the change of plant productivity affects the input of plant products (above- and belowground litter), causing changes in fractions of particulate organic carbon (Shi et al., 2024), subsequently affecting soil water-holding capacity and storage-discharge relationship. Furthermore, as topsoil texture becomes finer or coarser, water infiltrating into the deep soil changes, which in turn affects vegetation physiology and structure across scales (Wankmüller et al., 2024)."

3. The manuscript is not consistent in its portrayal of the method. Late in the manuscript (Section 4.2) the method is described as "a 'toy' model, also known as a minimal or exploratory model". The implication is that it is a first step and is intended to explore the possibilities of alternative methods, which seems appropriate. It also implies the approach has strong limitations. This contrasts sharply with the abstract and short summary, which seem to be saying that this is a method that can replace conventional hydrological models because it "excels at explaining pattern of slow hydrological behaviours" and the case study application "successfully captured slow hydrological behaviours". I strongly suggest that these matters be harmonised, preferably by changing the wording of the abstract and short summary and adding phrases such as "toy model" and "exploratory analysis". The abstract should also contain a clear statement of limitations.
Reply: Thank you for the comment. More explanations have been added in Abstract (line 37-38) and Short summary (line 46-47) for the revision.

"The system dynamics model is in its early stage with applications primarily confined to water-stressed regions and long-term scales."
"In spite of the simplicity, it holds potential to integrate hydrological behaviors across scales."

4. Following on from the above, the use of the method to provide projections (Fig. 7) is questionable. It is never explained why projections are provided in the first place. What use are projections produced by a "toy" model that is acknowledged to be exploratory? I suggest these matters be clarified.
Reply: Thank you for the comment. The projection serves as a form of model comparison aimed at validating our model's ability for long-term simulations, as initially we didn't conduct a comparative analysis between our model and a rainfall-runoff model. We believe that this comparison is not only more rigorous but

also offers additional information about future scenarios. A sentence has been added in line 395-397 for the revision.

"These findings further emphasized the system dynamics' structure is capable of producing long-term hydrological behaviors. Conversely, while adept at capturing short-term fluctuations, process-based models fall short in simulating long-term hydrological behaviors due to lack of the structure."

5. The wording of the title is not correct English. Two acceptable alternatives are:

- Can system dynamics explain long-term hydrological behaviours? The role of endogenous linking structure
- System dynamics explains long-term hydrological behaviours: the role of endogenous linking structure.

Given the issues raised above, it is more appropriate to phrase as a question - thus, I recommend the first one.
Reply: Thank you for the comment. The title has been changed accordingly.

6. Line 449-453. A correlation of X on a daily or monthly timestep has a different meaning to a correlation of X on an annual timestep. Thus, when comparing performance among methods, please state the timesteps on which the performance is calculated.
Reply: Thank you for the comment. The timesteps have been added in line 459-462 for the revision.

"Results showed that all models performed well in mean annual Q simulation but struggled with the simulation of inter-annual Q change, with median correlation coefficients ($r$) close to 0 for GCMs, and around 0.6 for GHMs and LSMs (r-square values ranging from 0.5-0.6) (Zhou et al., 2012; Hou et al., 2023)."

7. Line 467. The comments about an "attractor" seem to assume there is only one attractor. Some studies suggest there can be more than one - particularly Peterson et al. https://doi.org/10.1126/science.abd5085. Given the Peterson study was, like this study, framed around multi-annual behaviour in a relatively dry region, I suggest that the implications for the present manuscript be briefly discussed.
Reply: Thank you for the comment. A sentence has been added in line 497-498 for the revision.

"It is noteworthy that the ESMS is changeable in our study, as the factors that determine ESMS have evolved in tandem with the climate change, corresponding to multiple attractors."